# Phenotypic and Genomic Diversification in Complex Carbohydrate-Degrading Human Gut Bacteria

Nicholas A. Pudlo,[a] Karthik Urs,[a] Ryan Crawford,[b] Ali Pirani,[a] Todd Atherly,[c,d] Roberto Jimenez,[e] Nicolas Terrapon,[f,g] Bernard Henrissat,[f,g,h] Daniel Peterson,[e,i] Cherie Ziemer,[c,d] (ID) Evan Snitkin,[a] (ID) Eric C. Martens[a]

[a]Department of Microbiology and Immunology, University of Michigan Medical School, Ann Arbor, Michigan, USA
[b]Department of Computational Medicine and Bioinformatics, University of Michigan Medical School, Ann Arbor, Michigan, USA
[c]Iowa State University, Department of Animal Science, Ames, Iowa, USA
[d]United States Department of Agriculture Agricultural Research Station, Ames, Iowa, USA
[e]University of Nebraska, Department of Food Sciences, Lincoln, Nebraska, USA
[f]Aix Marseille Univ, CNRS, UMR7257 AFMB, Marseille, France
[g]INRAE, USC1408 AFMB, Marseille, France
[h]Department of Biological Sciences, King Abdulaziz University, Jeddah, Saudi Arabia
[i]Johns Hopkins University, Department of Pathology, Baltimore, Maryland, USA

Nicholas A. Pudlo and Karthik Urs contributed equally to this work. Author order was determined alphabetically.

**ABSTRACT** Symbiotic bacteria are responsible for the majority of complex carbohydrate digestion in the human colon. Since the identities and amounts of dietary polysaccharides directly impact the gut microbiota, determining which microorganisms consume specific nutrients is central for defining the relationship between diet and gut microbial ecology. Using a custom phenotyping array, we determined carbohydrate utilization profiles for 354 members of the *Bacteroidetes*, a dominant saccharolytic phylum. There was wide variation in the numbers and types of substrates degraded by individual bacteria, but phenotype-based clustering grouped members of the same species indicating that each species performs characteristic roles. The ability to utilize dietary polysaccharides and endogenous mucin glycans was negatively correlated, suggesting exclusion between these niches. By analyzing related *Bacteroides ovatus/Bacteroides xylanisolvens* strains that vary in their ability to utilize mucin glycans, we addressed whether gene clusters that confer this complex, multi-locus trait are being gained or lost in individual strains. Pangenome reconstruction of these strains revealed a remarkably mosaic architecture in which genes involved in polysaccharide metabolism are highly variable and bioinformatics data provide evidence of interspecies gene transfer that might explain this genomic heterogeneity. Global transcriptomic analyses suggest that the ability to utilize mucin has been lost in some lineages of *B. ovatus* and *B. xylanisolvens*, which harbor residual gene clusters that are involved in mucin utilization by strains that still actively express this phenotype. Our data provide insight into the breadth and complexity of carbohydrate metabolism in the microbiome and the underlying genomic events that shape these behaviors.

**IMPORTANCE** Nonharmful bacteria are the primary microbial symbionts that inhabit the human gastrointestinal tract. These bacteria play many beneficial roles and in some cases can modify disease states, making it important to understand which nutrients sustain specific lineages. This knowledge will in turn lead to strategies to intentionally manipulate the gut microbial ecosystem. We designed a scalable, high-throughput platform for measuring the ability of gut bacteria to utilize polysaccharides, of which many are derived from dietary fiber sources that can be manipulated easily. Our results provide paths to expand phenotypic surveys of more diverse gut

Address correspondence to Eric C. Martens, emartens@umich.edu.

The authors declare no conflict of interest.

bacteria to understand their functions and also to leverage dietary fibers to alter the physiology of the gut microbial community.

**KEYWORDS** *Bacteroides*, microbiome, pangenome, polysaccharides

Microbial communities in the distal intestines of humans and other mammals play critical roles in the digestion of dietary polysaccharides (1–3). Unlike proteins, lipids, and simple sugars, which can be assimilated in the small intestine, the vast majority of nonstarch polysaccharides (fibers) transit undegraded to the distal gut due to a lack of requisite enzymes encoded in the human genome (4). Microbial transformation of dietary fiber polysaccharides into host-absorbable organic and short-chain fatty acids is a beneficial process that unlocks otherwise unusable calories from our diet (5), shapes the composition and behavior of the gut microbial community (6–8), provides preferred nutrients directly to the colonic epithelium (9–11), and shapes the development of immune cell populations (12, 13).

The abundance of dietary fiber in the mammalian diet and the substantial chemical diversity within this class of molecules provide a prominent selective pressure that drives genome evolution and diversification within symbiotic bacterial populations. The genomes of individual human gut bacteria frequently encode dozens to hundreds more polysaccharide-degrading enzymes than humans secrete into the gastrointestinal tract, reflecting gut microbial adaptations to degrade dietary fibers (3, 4). As examples, the genomes of a few well-studied Gram-negative *Bacteroides* (*Bacteroides thetaiotaomicron*, *Bacteroides ovatus*, and *Bacteroides cellulosilyticus*) encode between 250 and over 400 CAZymes that collectively equip them to target nearly all commonly available dietary polysaccharides (14–16). However, none of these three species is by itself capable of degrading all available polysaccharides, a conclusion that was supported by early phenotypic surveys of cultured human gut bacteria that encompassed species from other phyla (17, 18). These findings suggest that individual microbes fill multiple, specific carbohydrate degradation niches and that a diverse community is required to ensure degradation of the entire repertoire of dietary fibers. Given that hundreds of different microbial species typically coexist in an individual over long time periods (19), it is important to understand how many different polysaccharide metabolism pathways are present within the individual microbial species that compose a community and how these traits are represented across strains and species. If some species possess very similar phenotypic abilities, they may be functional surrogates or compete for similar niches and therefore seldom co-occur.

Members of the *Bacteroidetes* phylum are often among the most numerous bacteria in the human colonic microbiota, with members of the genus *Bacteroides* often prominent in individuals from industrialized countries (19–21). These bacteria are well appreciated for their abilities to degrade a broad range of polysaccharides (16–18, 22, 23) and modify disease states in a bacterial species-specific fashion (24–26). In this study, we empirically measured the abilities of members of 29 different *Bacteroidales* species to grow on a custom panel of carbohydrates that span the diversity of plant, animal, and microbial polysaccharides. Our results reveal a wide range of metabolic breadth between different species, indicating that some have evolved to be carbohydrate generalists, while others have become metabolically specialized to target just one or a few nutrients. A pangenome analysis of several related strains provides insight into the evolutionary events that shape carbohydrate utilization among these important symbionts and reveals a dizzying mosaic of heterogeneity at the level of discrete gene clusters mediating polysaccharide metabolism. Based on the analysis of several variable loci, we provide evidence to support a mechanism of lateral gene transfer that may account for this mosaic architecture. Our results provide a glimpse into the metabolic breadth and diversity of an important group of human gut bacteria toward polysaccharide metabolism. Given the large amount of genomic and metagenomic sequence information that has been generated from the human microbiome, phenotypic studies

such as the one presented here represent important next steps in deciphering the functionality of these organisms in their native gut habitat.

## RESULTS

Phenotypes are the ultimate measures of biological function. However, large-scale phenotypic analyses are still uncommon in surveys of the human gut microbiome, which have instead relied on sequence-based approaches to infer function, often with substantial uncertainty. This lack of phenotypic information is due partly to a lack of high-density (e.g., strain level) culture representation for the dominant taxa combined with a lack of defined growth conditions to measure the behavior of these organisms. With the resurgence of gut microbial culturing, both of these gaps have begun to close (27–30), revealing an urgent need for scalable platforms to define the actual behavior of these organisms. To address this gap, we assembled a collection of human and animal gut *Bacteroidetes* and constructed a custom anaerobic phenotyping platform centered around carbohydrate metabolism, a key function that symbiotic gut microorganisms contribute to mammalian digestion (4). This array consists of 45 different carbohydrates (30 polysaccharides and 15 monosaccharides) that span the repertoire of common sugars and linkages present in dietary plants and meat, as well as host mucosal secretions and some rare nutrients consumed in regional populations or as food additives (see Fig. S1 in the supplemental material for a summary of polysaccharide structures).

The carbohydrate utilization abilities of 354 different human and animal *Bacteroidetes* strains were measured by individually inoculating each into this custom growth array and automatically monitoring anaerobic growth every 10 to 20 min for 4 days (see Materials and Methods). Based on the 16S rRNA gene sequence for each strain, this collection encompasses 29 different species based on the requirement that each strain possesses ≥98% 16S rRNA gene identity to a named type strain in a given species (Table S1a) (note that all but three strains, which were all related to each other and to *Bacteroides uniformis*, met this criterion). The resulting 31,860 individual growth curves were first inspected manually and then subjected to automated analysis to quantify total growth and growth rate parameters for each substrate (see Materials and Methods). A normalization scheme was employed to compensate for general growth differences in the two differently defined medium formulations employed (see Table S1a for a full list of strains assayed and all raw and normalized growth measurements; see Fig. S2 in the supplemental material for an analysis of replicates).

**Members of the same species possess similar carbohydrate utilization profiles.** Growth results are summarized in Fig. 1 and 2 and Fig. S3 in the supplemental material. Whether considered from the perspective of how many species degrade a particular polysaccharide (Fig. 1A) or how many individual polysaccharides are targeted by members of a particular species (Fig. 1B), there was substantial variability in carbohydrate utilization among the organisms surveyed (range, 1 to 28 polysaccharides degraded per strain; mean, 15.6). Some polysaccharides like soluble starch/glycogen were degraded by a majority of the species tested, and yet others like the edible seaweed polysaccharides carrageenan and porphyran were used by just one or two strains.

Given the diversity in observed carbohydrate utilization phenotypes, we wished to address if closely related strains display similar abilities or instead if strains of the same species have diverged from one another. To assist in visualizing the overall trends in carbohydrate utilization across this phylum, we performed unsupervised clustering of the strains based on their carbohydrate utilization profiles. While many species are not deeply represented by multiple strains, clustering based on a combination of normalized growth and rate measurements largely grouped strains of the same species together (Fig. 2), and as expected, this clustering was driven mostly by polysaccharide utilization abilities (see Fig. S4 in the supplemental material).

Our data reveal that strains belonging to several individual species possess more similar polysaccharide-degrading abilities to each other than their more distant

**A.**

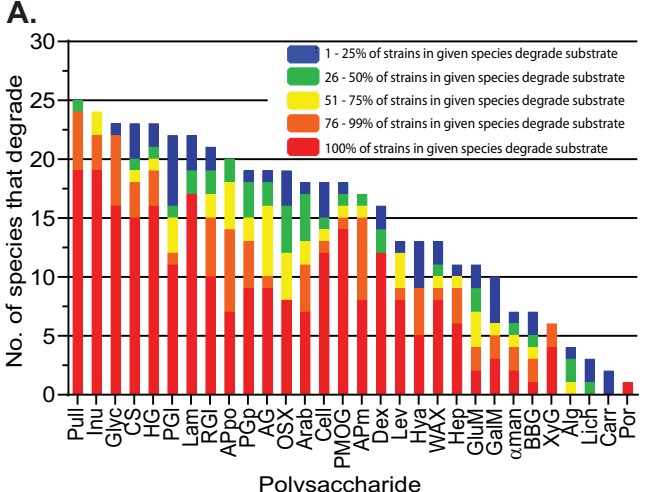

**B.**

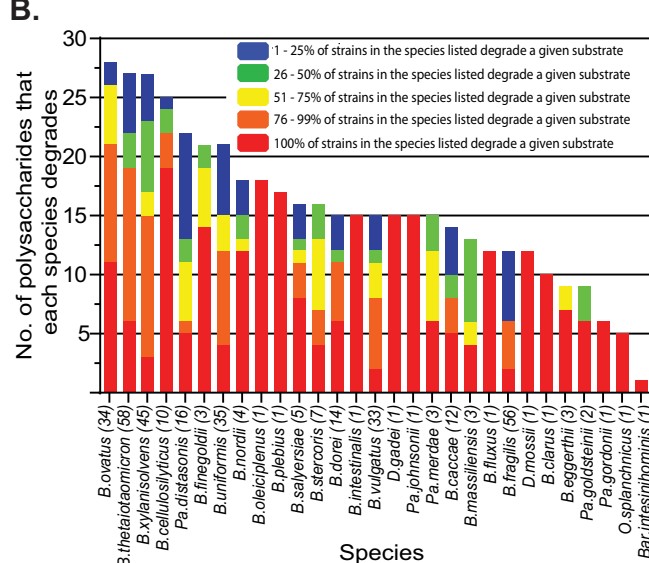

FIG 1 Glycan degradation abilities among gut *Bacteroidetes*. (A) The number of species out of 29 tested that degrade each polysaccharide is listed in order of decreasing degradation frequency from left to right. Since not all strains within a given species necessarily have the metabolic potential to utilize each polysaccharide, colors illustrate the percentage of strains within each degrading species that possess the indicated ability. (B) The number of polysaccharides that a given species degrades is shown in decreasing order. The number of strains tested for each species is listed in parentheses, and colors represent the percentage of strains in each indicated species that degrade each glycan counted toward the total.

relatives, a finding that has importance for interpreting or predicting function based on community sequencing data. As examples, all 56 strains of *B. fragilis* clustered together, reflecting their generally restricted abilities to utilize forms of soluble starch/ glycogen, inulin, and mucus *O*-glycans. Likewise, all 36 strains of *B. uniformis*, a species with a broader metabolic capacity that includes digestion of plant cell wall hemicelluloses, were also grouped together into a single branch. The inclusivity of these groupings was generally independent of the time period when strains were isolated or whether they were isolated from humans or other mammals (Fig. 2).

Another important feature of the observed species clustering is that the grouping does not mirror the overall phylogeny of the gut *Bacteroidetes*. Rather, phylogenetically separated species often group adjacent to one another based on similarities in carbohydrate metabolism (e.g., *B. ovatus/B. xylanisolvens* and *B. cellulosilyticus*, and *B. vulgatus/B. dorei* and *B. fragilis*) (see Fig. 3A for a phylogenetic tree based on conserved housekeeping genes) (31, 32). It is interesting to directly compare *B. fragilis* and *B. vulgatus/B. dorei*, which are two groups with deep strain representation (Fig. 2). Despite being phylogenetically more distant, members of these two species possess similar abilities to degrade starch and related molecules (glycogen and pullulan), inulin, and mucin *O*-glycans. The major distinguishing feature between these groups is the presence of some pectin utilization, which is often weak, among strains of *B. vulgatus/B. dorei*. Indeed, acquisition of growth abilities that are unique with respect to species with an otherwise similar potential may be one way that species avoid direct competition for the same niches.

Some polysaccharides, especially those present in the cell walls of dietary plants, occur in the same physical context and presumably traverse the gut together, potentially exerting selective pressure for bacteria to use them simultaneously. To test for the co-occurrence of different polysaccharide utilization abilities within the 354 individual strains, we calculated the pairwise correlations between the utilization of any two polysaccharides by the same strain (see Fig. S5 in the supplemental material). This test might reveal tendencies to coutilize different polysaccharides that are chemically different (positive correlation) or avoid using substrates from incompatible niches (negative correlation), if they exist. The presence of two different soluble starches (potato and maize amylopectin) and two starch-like glycans (glycogen and pullulan)

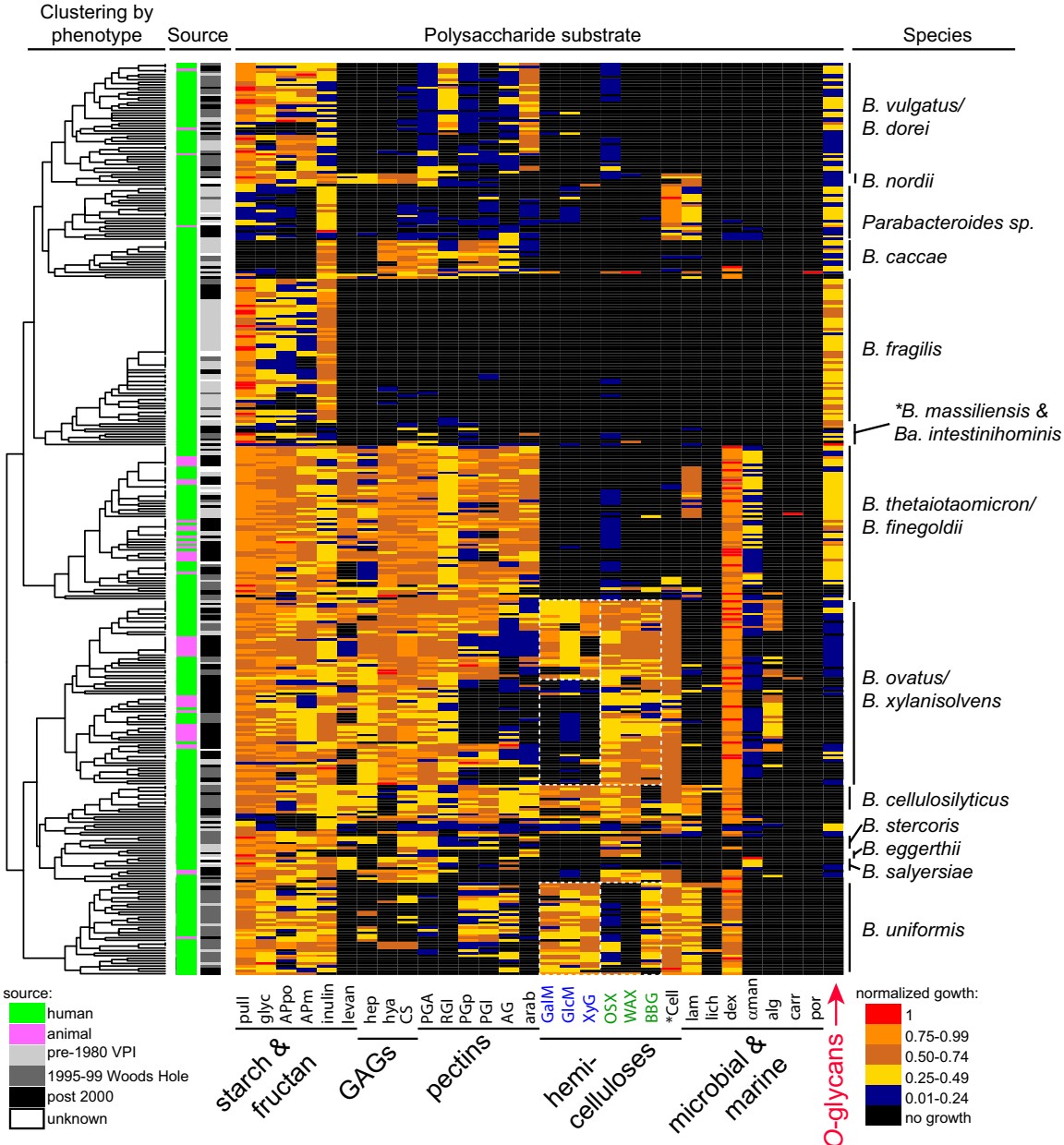

**FIG 2** Heatmap of individual polysaccharide utilization traits. Species are clustered by glycan utilization phenotype based on normalized total growth level (Fig. S4B). The magnitude of growth is indicated by the heatmap scale at the bottom right. Columns at the left indicate the source (human or animal) and time period of isolation. The cladogram at the far left shows the results of unsupervised clustering of the data based on the normalized growth data shown. The species designations at the right are the results of 16S rRNA gene sequencing (>98% identity to the species type strain was used to assign species). The region containing mucin specialists *B. massiliensis* and *B. intestinihominis* is indicated but marked with an asterisk because the 4 strains in these 2 species are not clustered perfectly in this region. All raw and normalized growth and rate data for individual strains may be found in Table S1. See Fig. S3 for an expanded heatmap with monosaccharide data and individual strain names labeled. All processed growth curves are available as source data.

provides an internal control since they are essentially identical in their sugar and linkage chemistry but vary in the proportion and placement of branches as well as polymer length, crystallinity, and solubility (Fig. S1). These four molecules are utilized through a single degradation/transport system in the type strain of *B. thetaiotaomicron*, which was included in our study (33). As expected, the abilities to use these four polysaccharides were among the strongest positive correlations (between 44% and 75%);

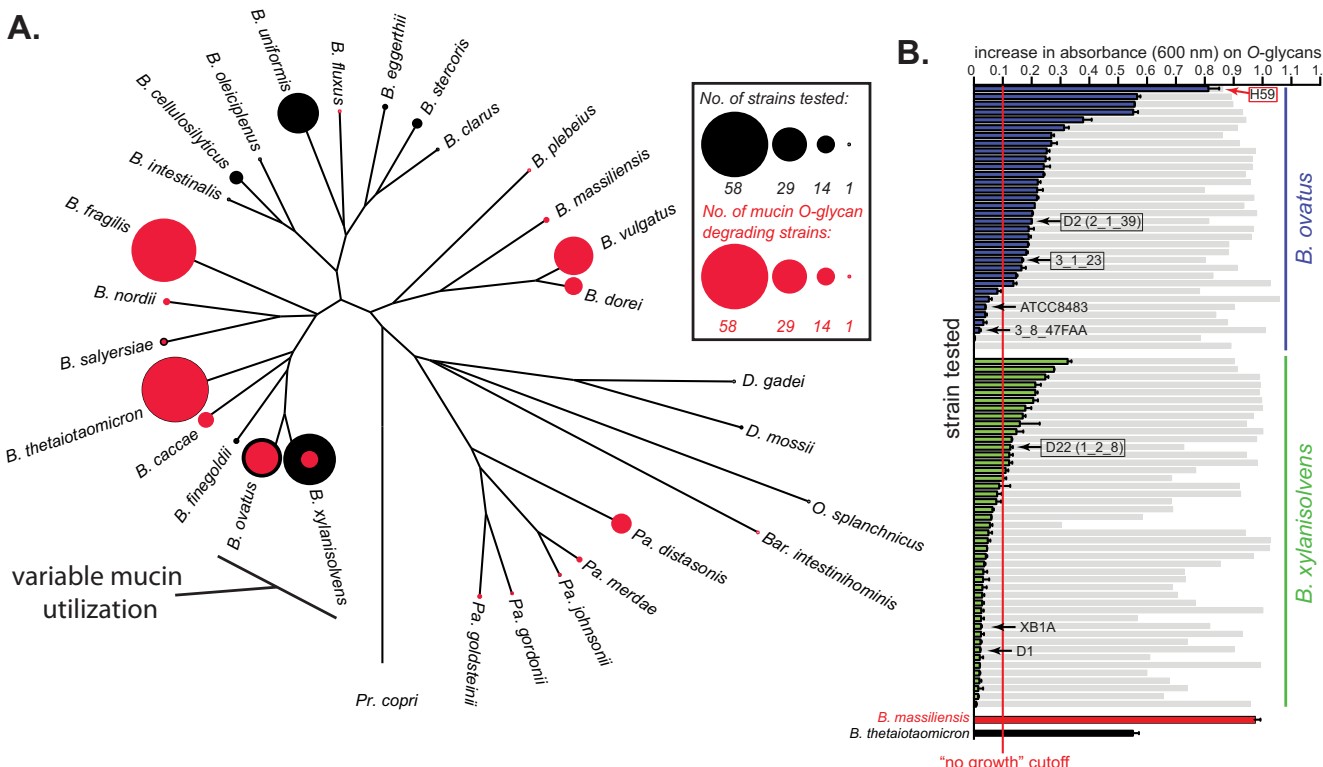

**FIG 3** Host mucin *O*-glycan metabolism within the *Bacteroides*. (A) A phylogenetic tree based on housekeeping genes that compares mucin *O*-glycan utilization across species. The diameter of the black circles represents the number of strains tested within each species (sample depth), whereas the size of the overlaid red circle corresponds to the number of strains exhibiting *O*-glycan metabolism. Note that some species have either full or no penetrance of this phenotypic trait and yet others like *B. ovatus*/*B. xylanisolvens* have more extensive variability among strains. (B) Strains of *B. ovatus* (blue) and *B. xylanisolvens* (green) that show variable growth abilities on mucin *O*-glycan (*n* = 2 growth assays per bar, error bars are range between values). Gray histogram bars are total growth controls on an aggregate of the monosaccharides that all strains of these two species grow on (Table S1) and are provided as a reference for overall growth ability on a non *O*-glycan substrate. Data from two established *O*-glycan degraders, namely, *B. massiliensis* and *B. thetaiotaomicron*, are also shown for reference. Species with black arrows were used for pangenome analyses to compare genetic traits associated with mucin *O*-glycan metabolism. We performed RNA-seq on three strains included in this pangenome analysis (black boxes) that were positive for *O*-glycan utilization and an additional strain, namely, *B. ovatus* NLAE-zl-H59 (red arrow, box), to see if there were unique genes/PULs present in strains that have the ability to grow on mucin *O*-glycans.

although, there was not a perfect correlation suggesting that some finer adaptation may exist even for different structural forms of a chemically similar molecule.

We also observed positive correlations in the ability of bacteria to simultaneously utilize polysaccharides within two different groups of plant cell wall polysaccharides (pectins and hemicelluloses), as well as animal tissue glycosaminoglycans (Fig. S5, green boxes highlight the 3 separate groups containing substrates with positive correlations within that group, although a weaker correlation can be observed across groups). These correlations occurred despite the fact that the polysaccharides within each of these groups often possess different structures but might co-occur in plant material or digested animal tissue. In the case of the hemicelluloses, there was even some apparent separation based on dicotyledonous versus monocotyledonous sources. The predominantly dicot hemicelluloses (Fig. 2, blue labels) and monocot hemicelluloses (Fig. 2, green labels) show some exclusivity with respect to the bacteria that utilize them. Many *B. ovatus*/*B. xylanisolvens* strains lack the ability to utilize the three dicot hemicelluloses (GalM, GlcM, and XyG), whereas the ability to degrade those from monocots (OSX, WAX, and BBG) is distributed more evenly. *B. uniformis* has a partially opposite pattern, preferring substrates from dicots, while only degrading one of the two major monocot structures (BBG) and poorly degrading the two xylans tested (OSX and WAX). Similar observations were also made for pectins and GAGs and could reflect adaptations to simultaneously harvest different nutrients from digesta particles derived from dicot plant cell walls or animal tissue ingested in a carnivorous diet. Finally, there

was a positive correlation between the utilization of $\alpha$-mannan and dextran, two microbial polysaccharides that are not known to occur together in foods or other sources of these polysaccharides (Fig. S5).

**Specialization for mucus *O*-linked glycans.** The most noteworthy correlation between polysaccharide utilization traits was observed between the utilization of host-produced mucin *O*-glycans and many of the other polysaccharides tested. Growth on a total of 19/30 polysaccharides showed negative correlations with the ability to utilize *O*-glycans, with the strongest negative correlations being between *O*-glycans and the seven different hemicelluloses (Fig. S5). This negative correlation is observed easily by comparing the rightmost column in Fig. 2 (*O*-glycan utilization) with the respective columns for hemicellulose degradation. Because this trend was observed across several species, it suggests that there could be a more general exclusive relationship between the two niches associated with foraging on mucus and hemicellulose. This idea is further supported by experiments described below, which suggest that isolates of *B. ovatus* and *B. xylanisolvens,* both adept hemicellulose consumers, are in the process of losing the ability to degrade *O*-glycans, relative to an ancestor that contained multiple gene clusters involved in the metabolism of these structures.

Interestingly, the mucin *O*-glycan mixture was the only substrate for which we observed absolute metabolic specialization among the substrates tested. A single, and only available strain of *Barnesiella intestinihominis* exhibited the ability to exclusively utilize mucin *O*-glycans, along with a subset of the sugars that are contained in these structures (Fig. 2; Table S1a). Three strains of *Bacteroides massiliensis* exhibited similar behavior with very strong growth on mucin *O*-glycans and only weak growth on soluble starches and a few other polysaccharides (Fig. 2; Table S1a). These three *B. massiliensis* strains were also restricted in the repertoire of simple sugars with which they could metabolize; this list is limited to those found in mucin and other host glycans (galactose, *N*-acetylgalactosamine, *N*-acetylglucosamine, *N*-acetylneuraminic acid, and L-fucose; weak fructose utilization by one strain was the only exception). Members of these two species are represented poorly in culture collections and remain lightly studied. However, their specific adaptations for host mucin glycans may render them important members of the microbiota, potentially thriving at the interface between the gut lumen and host tissue and relying exclusively on the host to be sustained. The continuous supply of mucin *in vivo* could explain why some species have become specialized for it as a nutrient, whereas dietary fiber degraders may need to be more generalist since the substrates available to them change with the host's meals.

**Pangenome reconstruction reveals extensive genetic diversification among related *Bacteroides* members.** With a view of the carbohydrate utilization traits present in our gut *Bacteroidetes* collection, we next sought to determine if certain variable traits were being gained or lost within strains of certain species and if available genomes provide insight into the mechanisms driving genomic adaptations to particular nutrients. Connections between polysaccharide utilization phenotypes and the underlying genes involved have been explored systematically for a few *Bacteroides* species (*B. thetaiotaomicron*, *B. ovatus*, and *B. cellulosilyticus*) with partial analyses in others (6, 16, 22, 23, 34–37). These studies have revealed that, in essentially all cases, the ability to degrade a particular polysaccharide is conferred by one or more clusters of coexpressed genes termed polysaccharide utilization loci (PULs) (38). PULs share defining features, such as genes encoding homologs of outer membrane TonB-dependent transporters (SusC-like), surface glycan-binding proteins (SGBPs; or SusD- and SusE/F-like), usually an associated sensor/transcriptional regulator, and one or more degradative CAZymes (glycoside hydrolase [GH], polysaccharide lyase [PL], and carbohydrate esterase [CE]), as well as other enzymes like sulfatases or proteases. Since the presence of one or more cognate PULs is required to utilize a given polysaccharide and these genes typically exhibit large increases in gene expression in response to their growth substrate, we rationalized that we could focus on traits that were variable in closely related strains and locate the associated PULs by transcriptomic analysis to gain insight into the basis of their acquisition or loss.

To test this hypothesis, we focused on members of two closely related species, *B. ovatus* and *B. xylanisolvens*, for which there is noticeable interstrain variation in their ability to use mucin *O*-glycans (Fig. 2 and 3). The investigation of these two species also benefits from substantial culture depth and many strains with available sequences. The *O*-glycans attached to mucins represent a diverse family of over one hundred different structures (39), albeit with common linkage patterns (Fig. S1). Correspondingly, the ability to utilize these glycans is a complex trait, involving the simultaneous expression of at least 6 to 13 different *O*-glycan-inducible PULs in *B. thetaiotaomicron*, *B. massiliensis*, *B. fragilis*, and *Bacteroides caccae* (6, 22, 35). Among the *B. ovatus* and *B. xylanisolvens* strains that surpassed the threshold for growth on *O*-glycans, there was a continuous gradient of growth abilities, which could be attributed to variations in PUL content and therefore gradations in the ability of the strains to access the many different structures in the complex *O*-glycan mixture (Fig. 3B). One hypothesis to explain this observation is that some *B. ovatus* and *B. xylanisolvens* strains have gained the ability to utilize *O*-glycans relative to an ancestor that lacked this phenotype. If so, the PULs they express during *O*-glycan degradation might be unique to their genomes and may indicate lateral gene transfer (LGT), as has been the case for the acquisition of phenotypes such as porphyran, agarose, and λ-carrageenan utilization in gut *Bacteroides*, which are all components of integrative conjugative elements or mobilizable plasmids (31, 40). An alternative hypothesis is that some *B. ovatus* and *B. xylanisolvens* strains are in the process of losing this ability from a common ancestor. If so, the genomes of nondegraders may still contain some PULs that are homologous to those present in more proficient *O*-glycan-degrading strains, but these strains may have lost a key step(s) that has eroded their ability to express this phenotype.

To distinguish these hypotheses, we selected seven strains (black arrows in Fig. 3B) that vary in their ability to degrade *O*-glycans and for which genome sequences exist. Note that three strains that degrade *O*-glycans were chosen initially because they were among the strongest degraders in our data set with sequenced genomes when we initiated these experiments. We later identified strains with better *O*-glycan growth abilities and address one of these (strain H59) separately below. Four of the selected strains were *B. ovatus* (two positive and two negative for *O*-glycan degradation); three strains were *B. xylanisolvens* (one weakly positive and two negative for *O*-glycan degradation). One of these strains (*B. xylanisolvens* XB1A) has a finished circular genome and was used as a scaffold to align the remaining six draft genome sequences, with manual curation (see Materials and Methods), resulting in a nearly contiguous pangenome sequence that captures the spatial arrangement of homologous and variable genes that are present in these seven strains (see Table S2a in the supplemental material) (see https://www.ericmartenslab.org/ for downloadable physical maps of the pangenome).

An analysis of the *B. ovatus*/*B. xylanisolvens* pangenome revealed remarkable variability in gene content among just the seven strains used. A total of 12,960 different genes were delineated based on ≥90% identity in their translated amino acid sequence (Table S2a). Remarkably, only 2,264 (17.5%) of these genes were shared among all 7 strains. The largest proportion of genes (7,244, 55.9%) was present only in 1 of the 7 strains. Separating two major classes of core PUL functions, SusC/D homologs and degradative CAZymes (GH, PL, and CE), revealed that these key components of *Bacteroidetes* polysaccharide metabolism were also represented heavily in the "accessory gene" pool that is not common to all strains (Fig. 4A).

Through informatics-based and manual annotation of gene clusters containing typical PUL functions, we delineated between 180 and 236 different PULs in the reconstructed pangenome (ambiguity is caused by many PULs occurring adjacent to each other; although in many cases separation of adjacent PULs according to individual genomes allowed us to make more precise delineations) (Table S2b). A direct comparison of the *O*-glycan-degrading and nondegrading strains revealed that there was a substantial number of genes (3,351) that were unique to the 3 *O*-glycan degrading

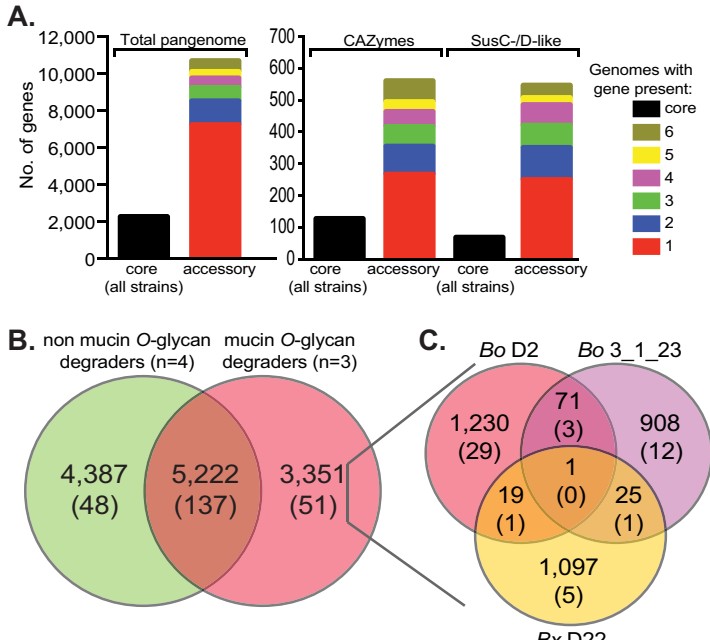

**FIG 4** Distribution of all genes as well as core polysaccharide utilization functions in the *B. ovatus/B. xylanisolvens* pangenome. (A) Left, shows the number of core genes (i.e., those present in all 7 strains used for pangenome construction) compared with genes present in 2 to 7 of the individual strains. Right, shows the same distribution of genes assigned to PULs or particular degradative CAZyme families (GH, PL, and CE) (see Tables S2 and S3 for more detailed assignments). (B) The distribution of genes between mucin-degrading (*n* = 3) and nondegrading (*n* = 4) strains used to construct the pangenome. Top numbers indicate total genes, while numbers in parentheses indicate the number of PULs (not individual PUL genes) in each category. (C) Distribution of the genes that are unique to the three mucin-degrading strains within each genome. Genes/PULs are numbered as described for panel B. Note that no PULs are shared by all three strains.

strains, including genes belonging to 51 PULs (Fig. 4B). However, such a distribution in gene content might be expected given the overall large proportion of noncore genes in these seven strains, and there was correspondingly no indication that all three *O*-glycan-degrading strains shared overlapping PULs with each other; no PULs were common to all three *O*-glycan degraders, and only five PULs were shared by any two strains (Fig. 4C). Considering that there are 51 total PULs that are unique to the mucin-degrading strains, if these strains have gained the ability to degrade *O*-glycans from an ancestral lineage that lacked this ability, it likely occurred by the acquisition of separate gene clusters. To more directly distinguish between the two hypotheses given above, we performed transcriptional profiling on all three *O*-glycan-degrading strains to determine if the PUL genes that they express during *O*-glycan degradation are indeed unique to these strains.

Compared with reference growth in minimal medium containing glucose (MM-glucose), the *B. xylanisolvens* D22, *B. ovatus* 3-1-23, and *B. ovatus* D2 strains activated the expression of 196, 227, and 359 total genes more than 10-fold, and these gene lists included components of 14, 19, and 42 different PULs, respectively (see Table S3a to c in the supplemental material). As expected from studies in other *Bacteroides*, these PULs were scattered throughout the genome (see Fig. S6 in the supplemental material), suggesting that they are regulated autonomously in response to glycan cues present in the *O*-glycan mixture. Strikingly, the majority of PULs that contained *O*-glycan-activated genes (63/75, 84%) were not unique to the *O*-glycan-degrading strains (Table S3a to c; Fig. S6). Moreover, in each of the three strains analyzed, the most highly upregulated PULs were also often shared with non-mucin-degrading strains. These observations lend support to the hypothesis that strains of *B. ovatus* and *B. xylanisolvens* are in the process of losing the ability to utilize *O*-glycans relative to a

common ancestor that possessed a more expansive gene repertoire to successfully access these nutrients. However, we cannot rule out that individual nondegrading strains are separately acquiring PULs that are associated with mucin degradation and retaining them without the full benefit that presumably occurs with the ability to fully execute this growth phenotype. This latter idea is consistent with interspecies PUL exchange observations elaborated below.

Finally, because we subsequently identified a *B. ovatus* strain (NLAE-zl-H59, red arrow in Fig. 3B) with a substantially higher ability to use *O*-glycans relative to the strains used for pangenome construction, we performed an additional transcriptome sequencing (RNA-seq) analysis on this strain. Compared with a glucose reference, this strain activated 373 total genes in response to *O*-glycans, including genes from 30 different PULs (Table S3d). Among these PULS, 26 activated PULs were also present in 1 of the 7 strains in our pangenome and 24 were homologous to PULs in strains that did not degrade *O*-glycans. However, this strain did activate the expression of genes within four PULs that were completely unique to its genome compared with the seven strains used for pangenome reconstruction, suggesting that it could possess additional genes that augment its ability to grow on mucin *O*-glycans. This increased PUL expression could be responsible for the enhanced growth of the H59 strain on *O*-glycans, especially if genes included within these unique PULs are responsible for key metabolic steps required for efficient *O*-glycan utilization.

**Evidence that intergenomic recombination has driven *Bacteroides* pangenome evolution.** Similar to other bacteria, we observed that many accessory genes in the *B. ovatus* and *B. xylanisolvens* pangenome are located in contiguous clusters or "islands," often involving PULs or capsular polysaccharide synthesis gene cluster (Table S2a). In contrast to previously identified *Bacteroides* PULs that have more obviously been subjects of lateral transfer (31, 40, 41) and are associated with integrative and conjugative elements (ICEs), most of the variable genomic regions that we identified were not associated with functions indicative of mobile DNA. Instead, these regions are often located precisely in between one or more core genes (i.e., those common to all seven strains; herein referred to as "genomic nodes") that flank each side of the variable gene segment (Fig. 5A and B).

Several intergenomic transfer mechanisms might account for the observed mosaic structure of the *B. ovatus-B. xylanisolvens* pangenome. The first is the movement of genes into a recipient genome by conjugation of mobile ICEs. While such events would be expected to leave behind residual genes involved in mobilization and transfer, which were not observed, these DNA vehicles are known to target a subset of core genes, such as tRNAs (41), and may have undergone subsequent genomic deletion events that eliminated the mobile DNA. Two other known mechanisms of bacterial LGT are natural competence and phage transduction, of which neither has been observed in members of *Bacteroidetes*.

A final potential mechanism is the direct conjugation of the chromosome from a donor bacterium into a related recipient, followed by subsequent homologous recombination between flanking nodes to add or delete intervening DNA in the recipient genome (Fig. 5C). This mechanism is conceptually similar to high-frequency recombination (Hfr) transfer in *Escherichia coli* and has already been described for *B. thetaiotaomicron* and *B. fragilis*. The mechanism involves chromosomal ICEs that may have lost their ability to circularize from the genome and instead act as transfer initiation points to conjugate a donor genome into a recipient, sometimes in response to the activity of other ICEs or conjugative transposons (42–44). If such a mechanism was active more broadly in LGT between *Bacteroides*, we would expect that some of the core/node genes involved would reflect sequence identities that were more similar to the donor bacterium from which they originated and this difference would be detectable more easily if the transfer was between members of different species like *B. ovatus* and *B. xylanisolvens*. Moreover, such transfer events could result either in the introduction of new genes into the recipient or elimination of genes depending on the genetic content in between recombination nodes from the donor chromosome.

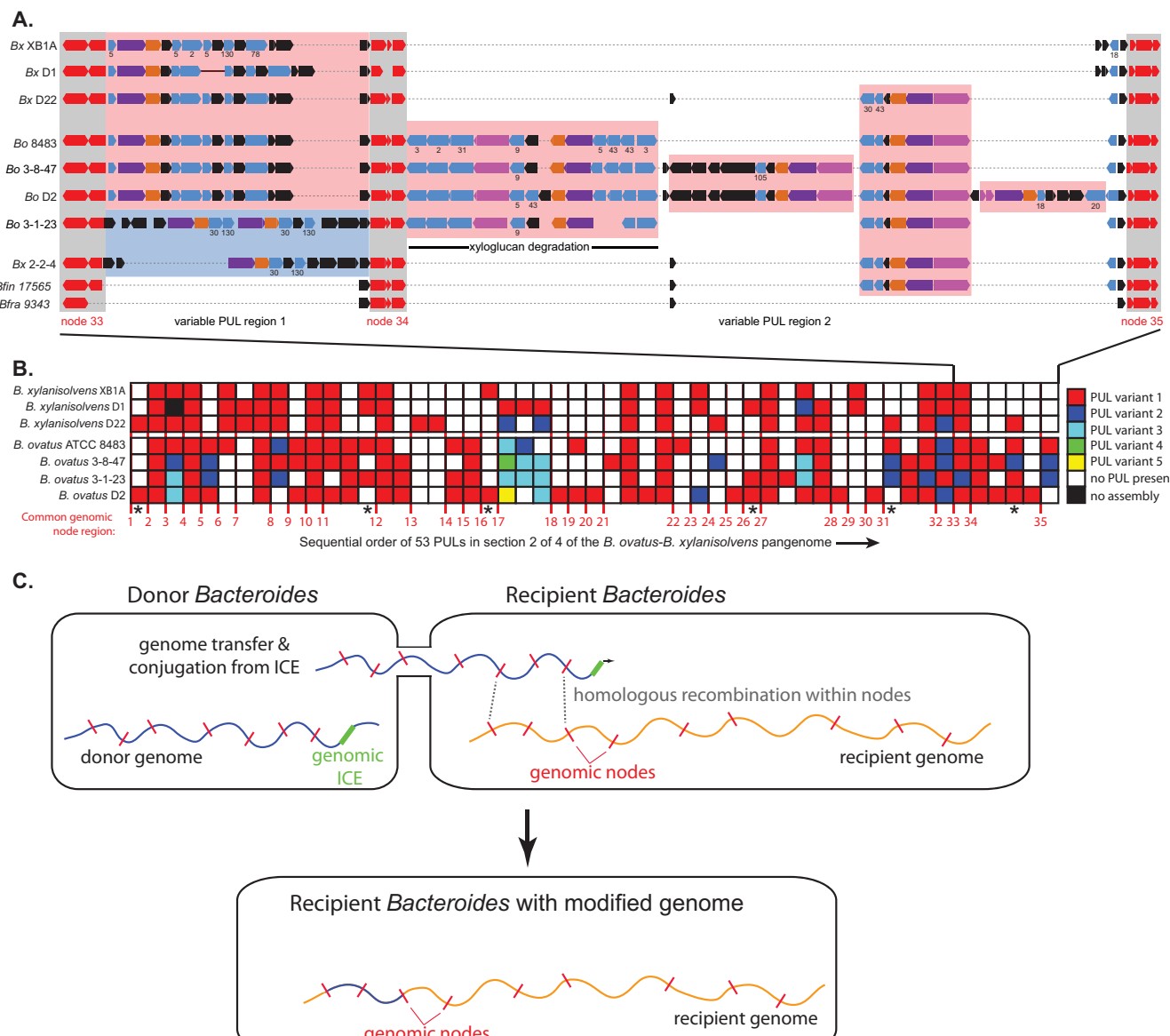

**FIG 5** Pangenome diversification in *B. ovatus* and *B. xylanisolvens*. (A) A higher-resolution view of a region of the *B. ovatus/B. xylanisolvens* pangenome shows the variable presence of at least 6 different PULs occurring between 3 genomic nodes (nodes 33 to 35 in this quarter of the total pangenome). Segment 2 of the physical pangenome map was selected because the first segment was initiated with numerous small contigs and this segment contained previously validated genes for xyloglucan metabolism (54). Node genes are colored red; while *susC*-like and *susD*-like genes are colored purple and orange, respectively; and glycoside hydrolase genes in light blue. GH family numbers are given below select PULs starting from the top to indicate potential specificity, and new numbers are only added going down the schematic if the family assignments are different, indicating a different PUL. A well-studied *B. ovatus* PUL for xyloglucan degradation (54) is shown in the center and occurs variably between two nodes and also has variable gene content. The two bottom genomes are from different species, namely, *Bacteroides finegoldii* (*Bfin*) and *Bacteroides fragilis* (*Bfra*) and show less complex genome architecture with the *Bacteroides fragilis* region possessing no PULs. (B) A broader view of the genome region in panel A, showing that the same mosaic pattern is common across the pangenome. Only PULs are illustrated, although many other genes were also variable in these regions. The numbers at the bottom delineate the presence of 35 different core gene nodes (as in panel A, some nodes contain multiple core genes) in this section of the genome, and the presence of homologous or unique PULs is illustrated according to the color code at right (see Fig. S6 for high-resolution physical maps of the pangenome with PUL annotations). Note that in some cases up to five different PULs were located at one location. (C) A schematic showing the proposed mechanism of genome exchange based on previous studies (42–44) and observations presented here. Genomic ICEs that are either partially active (excision deficient but capable of initiating DNA strand breakage and conjugation) or activated in *trans* by the presence of an exogenous conjugative transposon initiate genome mobilization from a donor into a recipient. If sufficient homology between node genes exists in the recipient, homologous recombination between two nodes can replace a section of the recipient with a segment from the donor. Note that genomic regions are shown as linear fragments for simplicity but would be circular.

To test this hypothesis, we took a bioinformatics approach aimed at first identifying high-confidence examples of interspecies recombination involving core genes and then assessed whether those genes were associated with the cotransfer of adjacent or intervening accessory genes (Fig. 6A). We collected a data set of 33 *B. ovatus* and

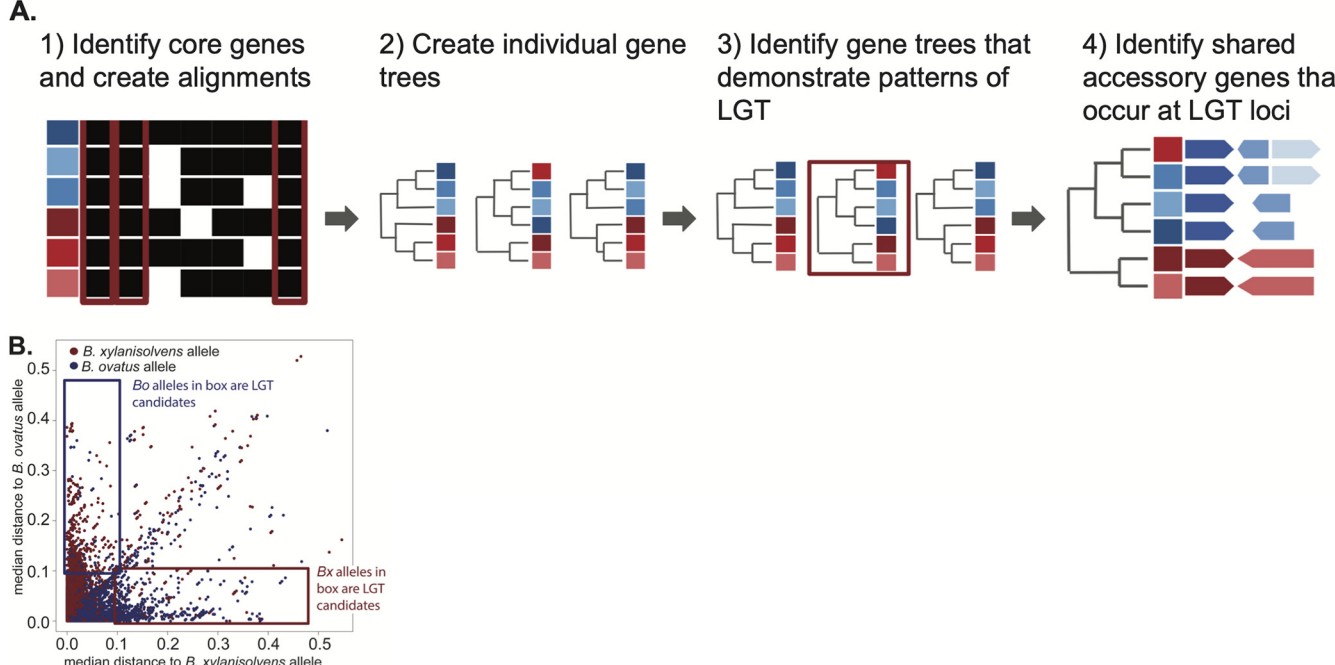

**FIG 6** Identification of putative lateral gene transfer events. (A) Schematic of the workflow to identify putative LGT core genes, which is described as follows: align genes and build corresponding trees for each core gene, determine the median substitution distances for each allele of a core gene in a given strain to both species, and identify loci with an identical conserved structure between isolates of opposite species. (B) Plot of median distances for all core genes identified in the 33 genomes analyzed. The boxes show the regions containing genes for which the median distance was >0.1 to the assigned species for a given strain and ≤0.1 for the opposite species to which a strain is assigned. These genes were determined to be high-confidence examples of core/node genes that had been replaced by an allele from the other species.

*B. xylanisolvens* genomes, which represent a subsample of the isolates for which we generated phenotypic data. We identified a set of 1,384 core genes—expectedly smaller than the core genome of the 7 strains used above due to additional strains being added—that are present as a single copy in all members of both species. To identify cases of putative interspecies LGT via homologous recombination at core genes, we searched for instances in which a core gene sequence from either species was more similar to the corresponding gene in the other species. We calculated the median distance of each strain-specific core gene to all other alleles of that core gene in strains belonging to both species (Fig. 6B, blue and red boxes indicate the core genes that are more similar to alleles in the other species). Among these candidate LGT genes/loci, we then investigated if any of these putative transfer events have resulted in pangenome diversification by searching for the presence of any accessory gene(s) that was observed only adjacent to a core gene with evidence of LGT.

In total, we identified 29 different loci at which the exchange of core genes appeared to have occurred and adjacent accessory genes were identified, including 7 that appeared to involve the transfer of PULs (Fig. 7A, see Fig. S8 in the supplemental material). Similar numbers of potentially transferred loci were identified for each species (16 loci in *B. xylanisolvens* and 13 loci in *B. ovatus*). Among the candidate LGT events, variable numbers of accessory genes were found within the loci ranging from 1 to 13 genes (Fig. 7A, Fig. S8). More genes (57 total) appeared to be transferred into *B. ovatus* than into *B. xylanisolvens* (36 total).

Finally, we determined if any of the identified LGT events could explain differential phenotypes measured by our high-throughput growth assay by modifying the complement of PULs in individual genomes. As a specific example, we focused on a PUL that was associated previously with *β*-mannan degradation (23, 45) that was among our candidate loci with evidence of transfer from a *B. xylanisolvens* ancestor into two *B. ovatus* strains. The presence of this PUL (PUL-A in Fig. 7A and B) was observed in all strains with the ability to grow on the *β*-mannan galactomannan (GalM), including

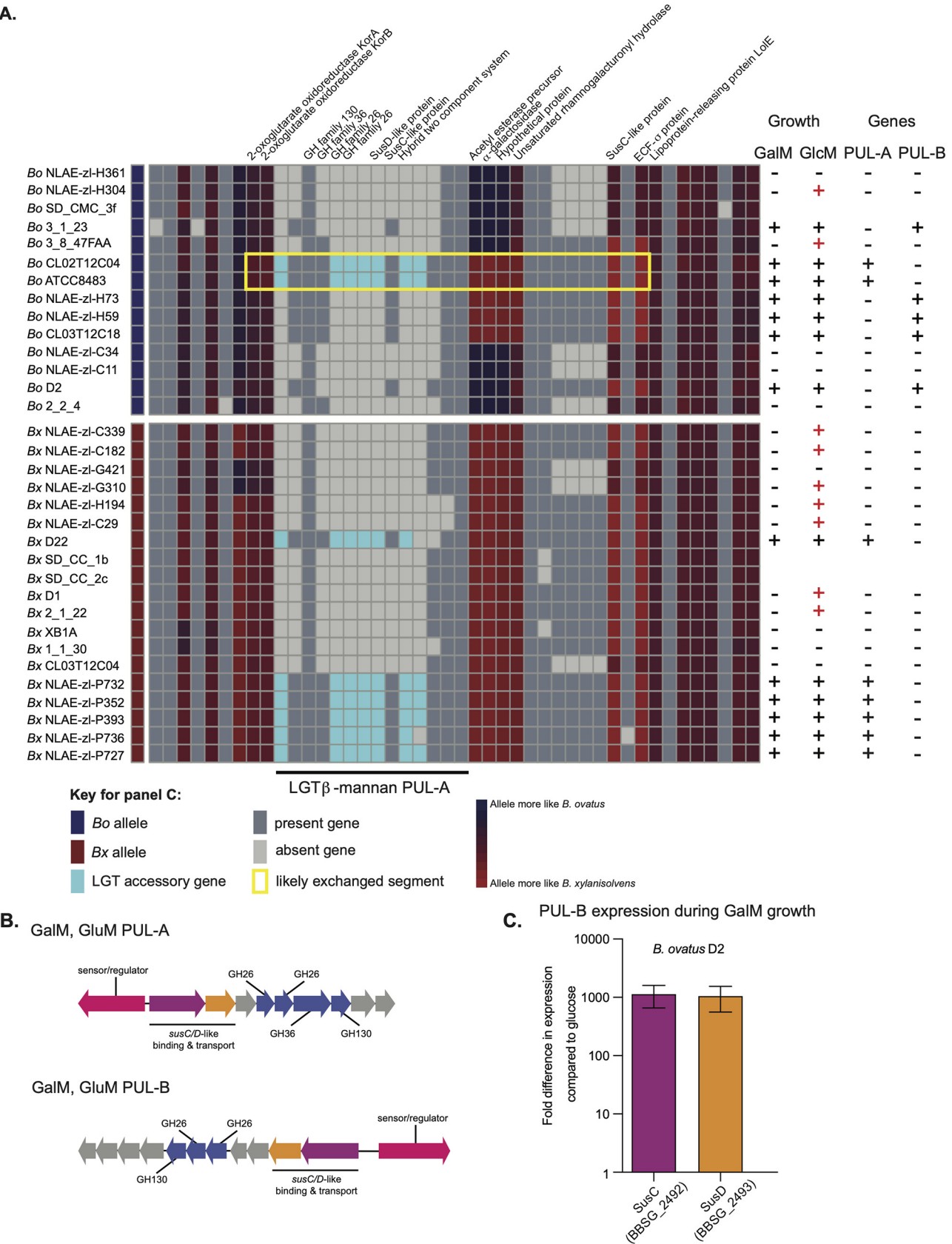

**FIG 7** Evidence that a PUL for β-mannan metabolism has been laterally transferred into *B. ovatus*. (A) A region of the *B. ovatus/B. xylanisolvens* pangenome that contains a PUL involved in galactomannan (GalM) and glucomannan (GluM) degradation. This PUL is present in

two strains of *B. ovatus* (ATCC 8483 and CL02T12C04) for which the flanking node regions were more similar to *B. xylanisolvens*. We showed previously that the deletion of this PUL from *B. ovatus* ATCC 8483 eliminated growth on GalM and glucomannan (GluM) (45), suggesting that it was both acquired from a *B. xylanisolvens* strain and conferred growth on these two β-mannans. However, the presence of this PUL was not correlated perfectly with growth on GalM, and several strains that lacked PUL-A still exhibited robust growth. Thus, we searched for other PULs that harbor GH26 family enzymes and determined that all of the other strains that grow on GalM, but lack PUL-A, harbor another candidate GalM PUL (PUL-B, Fig. 7B) at a different genomic location and some strains possess both (Fig. 7A). Gene expression analysis by quantitative PCR (qPCR) revealed that PUL-B was expressed highly in strains that lacked PUL-A during growth in GalM (Fig. 7C) and every strain that grew robustly on GalM had at least one of these two PULs. While we had previously shown that PUL-A was required for GlcM growth in *B. ovatus* ATCC 8483, there were a number of other strains (red "+" symbols in Fig. 7A) that displayed a weaker ability to grow only on GlcM, while lacking both of the GalM-associated PULs, suggesting the presence of additional PULs that confer the ability to grow on variant β-mannans. Such a presence of multiple nonorthologous PULs that confer the same or similar functions, and some which may be moving between genomes of related species by the putative LGT mechanisms noted above, complicates the process of understanding the genotype-phenotype relationships in human gut *Bacteroidetes* but will need to be resolved to make better functional predictions from sequence-based data.

## DISCUSSION

In this study, we leveraged a scalable, high-throughput quantitative growth platform to characterize the phenotypic abilities that are present in a sample of hundreds of *Bacteroidetes* strains from the human and animal gut. Our anaerobic screening technique is directly applicable to other bacterial phyla from the human gut and other environments. Moreover, it can be adapted to include new polysaccharides or to focus on different nutrient utilization or chemical resistance phenotypes. The current study, in concert with future applications of phenotypic screening, will help close the gap between our largely sequence-based view of the human gut microbiota and the functions that its members provide. However, instances like the ones investigated here for mucin glycan and β-mannan utilization by *Bacteroides* serve as a warning that the presence or absence of genes that are associated experimentally with a particular function do not always indicate that the phenotype is expressed or not.

Pangenome reconstruction for *B. ovatus* and *B. xylanisolvens* revealed extensive variability between strains of these closely related species, which is not unexpected for bacteria that engage in LGT. However, the lack of mobile DNA signatures for the majority of accessory genes and evidence of intergenomic recombination between species at core genes provide new insight into what may be a prominent mechanism of genome diversification in members of this phylum. The previously described intergenomic transfer mechanisms in *B. thetaiotaomicron* and *B. fragilis* required the presence of active or inactive ICEs, highlighting the potential roles for these mobile elements in not just shaping genomes directly but also indirectly through their ability to catalyze the exchange of broader genomic segments. In *B. thetaiotaomicron*, genome transfer was determined to initiate at genomically integrated ICEs of which there are four in the type strain of *B. thetaiotaomicron* (VPI-5482). They have not been shown to be fully

**FIG 7** Legend (Continued)
six strains of *B. xylanisolvens* and two strains of *B. ovatus*, and in the latter cases, flanking node genes exhibit signatures of being derived from LGT with a *B. xylanisolvens* donor (the yellow box highlights a potential recombination region). The columns at the left indicate the growth of each strain on GalM or GluM. The ability to grow on GalM is correlated fully with the presence of one of two different PULs, or both, that are transcriptionally activated during growth on this substrate (23). Notably, some strains (red "+") are able to grow weakly on GluM but do not possess either of the identified PULs, suggesting that additional, partially orthologous PULs exist that confer the ability to use only GluM. (B) Schematics of PUL-A and PUL-B associated with GalM and GlcM utilization. In *B. ovatus* ATCC 8384, elimination of PUL-A eliminates both of these growth abilities. (C) Expression analysis by qPCR of two sentinel genes from PUL-B in *B. ovatus* strain D2 that lacks PUL-A but still exhibits robust growth on GalM.

functional for circularization and mobilization. However, the introduction and activation of an additional, excision-proficient conjugative transposon (either cTnDOT or cTnERL) (42), which shares common features with the genomic ICEs, catalyzed the expression of genes in the genomic ICEs and transfer of parts of the genome in a manner that requires *recA* and homologous DNA to be present in the recipient (42). An additional study in *B. fragilis* showed that conjugation from a strain with multiple genomic ICEs, with one or more presumably retaining transfer activity, results in the transfer of up to 435 Kb of chromosome into a recipient that initiates near genomic ICEs, with individual transfer events being of variable size. The latter observation suggests that intergenomic recombination could then occur at different homologous regions (i.e., the core gene nodes observed in the pangenome), which could depend on the amount of genomic DNA transferred and the length/homology of available recombination sites. Given that the number of ICEs in individual genomes is variable and their ability to be activated by functional conjugative transposons that are circulating in the ecosystem may also vary, it will be interesting to determine in future work if there are hot spots for genome transfer or if certain strains/species are dominant genome donors that could play a disproportionate role.

The phenotypic similarity between members of the same species (e.g., *B. ovatus* and *B. xylanisolvens*) and the large amount of gene diversity, including genes involved in carbohydrate metabolism, present a paradox and raise the question of why the genome diversification observed in strains of *B. ovatus* and *B. xylanisolvens* has not pushed members of these species to behave more differently and cluster based on phenotype with members of other species. One answer may be the apparent exclusion of some traits, such as mucin *O*-glycan/hemicellulose metabolism, which may limit the fitness advantage associated with acquiring new phenotypes. A second emerges from the proposed genome-exchange mechanism for which we offer new bioinformatics support. Since this intergenomic exchange relies on homologous recombination, its frequency should decrease between genomes that are more divergent. Thus, this strategy may be one mechanism through which only closely related bacteria can share traits that are advantageous with other close relatives. The presence of nonorthologous PULs that confer the same function (e.g., GluM and GalM utilization), of which some appear to be subjected to LGT, further complicates interpretations of genotype-to-phenotype relationships in these bacteria. Based on the prevalence data, it seems that PUL-A is a GalMan utilization system that is more prevalent in, and perhaps also originated in, *B. xylanisolvens*, and it is also capable of transfer to *B. ovatus*. PUL-B is more prevalent in *B. ovatus* and may have origins in that species, at least with respect to *B. xylanisolvens* where it has so far not been observed. Notably, the genome transfer mechanism proposed here does not account for how new genes can be incorporated between conserved nodes. Rather, this variability must pre-exist among different strains and therefore be created by different inter- and intragenomic diversification mechanisms. Nevertheless, the data that we report here underscore the notion that individual gut symbiont genomes are not just highly variable but also dynamically so.

## MATERIALS AND METHODS

**Bacterial strains and growth conditions.** A total of 354 human and animal gut *Bacteroidetes* were included in this study. A complete list is provided in Table S1b, along with species designation based on 16S rRNA gene sequencing and associated metadata. Abigail Salyers (University of Illinois, Urbana-Champagne) kindly provided many of the strains, and 2 large portions of this collection were isolated over several decades, as follows: 99 strains with "WH" designations were collected from fecal samples of healthy human volunteers as part of the Woods Hole Summer Course on Microbial Diversity in the late 1990s, and 95 additional strains with "VPI" designations were collected from human samples at the Virginia Polytechnic Institute in the 1960s to 1970s. Species classifications were made based on alignment of a minimum of 734 bp of 16S rRNA gene sequence to a database containing the type strains of >29 named human gut *Bacteroidetes* species using the classify.seqs command with Bayesian settings in the program mothur (46); assignment for each strain was also checked manually by BLAST (47). Isolates with ≥98% 16 rRNA gene sequence identity to the type strain of a named species were labeled with that species designation. This classification strategy included all except for 3 of the 354 strains examined, which ranged between 96.6% and 96.7% sequence identity to the *B. uniformis* ATCC type strains, and

based on sequential isolate numbers might be clones from the same individual (see WH15, WH16, and WH17 entries in Table S1a). Because of the small number of strains that did not satisfy our 98% cutoff, we grouped these unclassified strains with their nearest relative and labeled them as more divergent in Table S1a; although, in most cases, the carbohydrate phenotypes of these strains were very similar to other members of the *B. uniformis* group.

All strains were grown routinely in an anaerobic chamber (Coy Lab Products, Grass Lake, MI) at 37°C under an atmosphere of 5% $H_2$, 5% $CO_2$, and 90% $N_2$ on brain heart infusion (BHI; Beckton Dickinson) agar that included 10% defibrinated horse blood (Colorado Serum Co.) and gentamicin (200 $\mu$g/mL). A single colony was picked into either tryptone-yeast extract-glucose (TYG) media (48) or modified chopped-meat carbohydrate broth (Table S1b) and then subcultured into a minimal medium (MM) formulation that contained a mixture of monosaccharides, vitamins, nucleotides, amino acids, and trace minerals (Table S1b provides components and a complete recipe).

**Carbohydrate growth array setup and data collection.** Two different minimal medium formulations were used in the carbohydrate growth arrays (Table S1a lists the formulation used for each isolate). The simpler of the two formulations (medium 1) was identical to the above MM, except that no carbohydrates were included and the medium was prepared at a 2× concentration. The second minimal medium formulation (medium 2) was identical to medium 1 but included beef extract (0.5% [wt/vol] final concentration) as an additional supplement. We initially attempted to cultivate all of the species tested using only medium 1 but determined that beef extract was specifically required to allow the growth of some species, especially *Parabacteroides* spp., *Barnesiella intestinihominis*, *Odoribacter splanchnicus*, and the branch of *Bacteroides* that includes *Bacteroides plebeius* and *B. massiliensis*. Growth in the absence of an added carbohydrate source was generally not observed or very low, except with *Parabacteroides* that were often able to grow to a low level on the added 0.5% beef extract. The corresponding negative-control wells for each strain assayed were averaged, and this value was subtracted from the total growth calculation of the corresponding to strain on other carbohydrates tested (all raw growth curves are provided as source data). Despite several attempts to supplement minimal media with different components or employ more stringent anaerobic methods, we were unable to cultivate several common *Bacteroidetes* genera/species (*Prevotella* spp., *Paraprevotella* spp., *Alistipes* spp., and *Bacteroides coprocola* and *Bacteroides coprophilus*) in these two MM formulations and therefore did not include them in this study. All of these isolates grew readily in rich medium, suggesting that they have specific nutritional requirements that were not met in the MM formulations used.

Carbohydrate growth arrays were run as described previously (23) using a list of 45 carbohydrates (see reference 23 for a complete list with supplier information) that were present in duplicate, nonadjacent wells of a 96-well plate; 2 additional wells contained no carbohydrate and served as negative controls. Each MM was prepared as a 2× concentrated stock without carbohydrates (MM-no carb). An aliquot of each strain was taken from a MM-monosaccharides culture (grown for 16 to 20 h) and was centrifuged to pellet cells. Bacteria were resuspended in the same volume of 2× MM-no carb and then centrifuged again prior to suspension in a volume of 2× MM-no carb that was equal to the original volume. These washed bacterial cells were then inoculated at a 1:50 ratio into 2× MM-no carb, and the suspension was added in equal volume (100 $\mu$L/well) to the 96 wells of the carbohydrate growth array. Each well of the carbohydrate growth array contained 100 $\mu$L of 2× carbohydrate stock (10 to 20 mg/mL); thus, when diluted 2-fold, it resulted in 1× MM containing a unique carbohydrate and a bacterial inoculum that was identical to other wells. Growth arrays were monitored at kinetic intervals of 10 to 20 minutes using a microplate stacking device and coupled absorbance reader (Biotek Instruments, Winooski, VT), and data were recorded for 4 d (variable kinetic interval times reflect variations in the number of microtiter plates present in a given batch).

**Carbohydrate growth array data processing.** Growth data were processed according to the following workflow: (i) data for each strain were exported from Gen5 software (Biotek Instruments, Winooski, VT) into Microsoft Excel and a previously described automated script was employed to call the points at which growth began (min) and ended (max) (23), (ii) each file was checked manually to validate that appropriate calls were made and the min and max values edited if needed (generally, only due to obvious baselining artifacts or erroneously high calls caused by temporary bubbles or precipitation); (iii) "total growth" ($A_{600}$ max − $A_{600}$ min) and "growth rate" [($A_{600}$ max − $A_{600}$ min)/($t$ max − $t$ min)] were calculated for each strain on each substrate ($A_{600}$ is the absorbance value at 600 nm that corresponds to each min and max point; $t$ is the corresponding time values in minutes; when necessary, the growth level associated with the average negative-control growth was subtracted from the total growth value), and (iv) individual cultures in which total growth was ≤0.1 were scored as "no growth" and their $A_{600}$ values converted to 0. Only assays in which both replicates showed an increase in $A_{600}$ of ≥0.1 were considered growth; if the 2 replicate assays were discordant (one positive, one negative), then both values were converted to 0.

To normalize the results for each strain, the substrate(s) that provided maximum total growth and growth rate values were determined, and they were set to 1.0. All other growth values for a given strain were normalized to this maximum value, providing a range of values between 0 and 1.0. We next normalized growth ability across individual substrates using the previously normalized values for each individual strain; the strain with the maximum total growth and growth rate values were identified (many of these were already set to 1.0). Then, the corresponding values for each other species on that particular substrate were calculated as a fraction of the maximum value for that substrate, yielding a range of values between 0 and 1.0 for each substrate. These values were used to create the heat map shown in Fig. 2 and Fig. S3, and all raw and normalized values are provided in Table S1a.

**Data clustering and statistics.** Heatmaps and corresponding dendrograms were generated using the "heatmap" function in the "stats" package of R (version 3.4.0) which employs unsupervised hierarchical clustering (complete linkage method) to group similar carbohydrate growth profiles. Pearson correlation was used to calculate the co-occurrence of the ability to grow on each pair of different substrates. The normalized growth value for each substrate was compared with the corresponding growth values on all other substrates using the Pearson correlation test in R, and these values are displayed in the Pearson correlation plot in Fig. S5.

**Pangenome reconstruction for *B. ovatus* and *B. xylanisolvens* strains.** Since one of the seven strains used for pangenome reconstruction (*B. xylanisolvens* XB1A) was assembled into a single circular chromosome, we used this genome as a scaffold for the contigs representing the remaining six strains. Contigs from the six unfinished strains were aligned against the XB1A genome using a combination of Mauve (49), to align and orient larger contigs, and reciprocal best BLAST-hit analysis using ≥90% amino acid identity to identify likely homologs, to provide finer resolution. Contigs from draft genome assemblies or *B. xylanisolvens* XB1A were broken as needed to accommodate the inclusion of unique accessory genes but only in circumstances where genes on both sides of the break could be aligned to homologs in one or more genomes with a contig that spanned that breakpoint. After constructing a preliminary assembly, we analyzed the size distribution of putative homologous open reading frames (ORFs) as a measure of assembly accuracy and to identify variations in genetic organization that might be attributable to real genetic differences such as frame shifts, which would result in two homologous gene calls of smaller size in the genome containing the frameshift. Any variation in >50% of homologous ORF size was inspected manually using the "orthologous neighborhood viewer, by best BLAST hit" function in the U.S. Dept. of Energy Integrated Microbial Genomes (IMG) website. Introduced contig breaks are documented in Table S2a. GenVision software (DNAstar, Madison, WI) was used to visualize and label selected functions in the pangenome assembly and also display RNA-seq data as a function of shared and unique PULs. Downloadable physical maps of the reconstructed pangenome are provided online at https://www.ericmartenslab.org/people.

**RNA-seq analysis.** For RNA-seq, *B. xylanisolvens* and *B. ovatus* cells were grown to mid-exponential phase on either purified mucin *O*-linked glycans (purified in-house from Sigma type III porcine gastric mucin) or glucose as a reference as previously described (22). Total RNA was extracted using an RNeasy kit (Qiagen) and treated with Turbo DNase I (Ambion), and mRNA was enriched using the bacterial Ribo-Zero rRNA removal kit (Epicentre). Residual mRNA was converted to sequencing libraries using TruSeq barcoded adaptors (Illumina) and sequenced at the University of Michigan Sequencing Core in an Illumina HiSeq instrument with 24 samples multiplexed per lane. Barcoded data were demultiplexed and analyzed using the Arraystar software package with Qseq (DNAstar). All RNA-seq data are available publicly from the National Institutes of Health Gene Expression Omnibus Database under accession numbers GSM4714867 to GSM4714890.

**Core gene determination and detection of LGT events between *B. ovatus* and *B. xylanisolvens* strains.** The core gene alignment was generated with cognac (50). The alignment was then partitioned into the individual component genes, and approximate maximum likelihood gene trees were generated with FastTree (51). Cophylogenetic distances were calculated with ape (52). A distance threshold of greater than 0.1 to the same species and less than 0.1 to the opposite species was used to identify alleles bearing signatures of HGT. All analyses were performed in R (version 3.6.3) (53). All code developed for this project are available online at https://github.com/rdcrawford/bacteroides_hgt.

## SUPPLEMENTAL MATERIAL

Supplemental material is available online only.

**FIG S1**, PDF file, 0.5 MB.
**FIG S2**, PDF file, 0.5 MB.
**FIG S3**, PDF file, 1 MB.
**FIG S4**, PDF file, 0.5 MB.
**FIG S5**, PDF file, 1.3 MB.
**FIG S6**, PDF file, 2.2 MB.
**FIG S7**, PDF file, 2.1 MB.
**TABLE S1**, XLSX file, 1.8 MB.
**TABLE S2**, XLSX file, 1.7 MB.
**TABLE S3**, XLSX file, 0.3 MB.

## ACKNOWLEDGMENTS

This work is dedicated to the memory of Cherie Ziemer.

We thank Thomas Schmidt (University of Michigan) for helpful advice on developing the phenotype clustering score. We thank Abigail Salyers (University of Illinois, Urbana-Champagne) who kindly provided a large portion of the strains used in this work. Additional support with strain culture and resources was provided by Nadja Shoemaker (University of Illinois, Urbana-Champagne), Emma Allen-Vercoe (University of Guelph), Laurie Comstock (Harvard University), Jin-Woo Bae (Kyung-Hee University, South

Korea), Tomomi Kuwahara (Kagawa University, Japan), and Jeffrey Gordon (Washington University).

This work was supported by funds from the U.S. National Institutes of Health (DK084214, DK118024, and DK125445), the University of Michigan Biological Sciences Scholars Program, and pilot/feasibility grants from the University of Michigan Center for Gastrointestinal Research (NIDDK DK034933).

We declare no competing interests.

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
