## [Reviewer comments · mSystems]

Phenotypic and genomic diversification in complex carbohydrate degrading human gut bacteria

Eric Martens, Nicholas Pudlo, Karthik Urs, Ryan Crawford, Ali Pirani, Todd Atherly, Roberto Jimenez, Nicolas Terrapon, Bernard Henrissat, Daniel Peterson, Cherie Ziemer, and Evan Snitkin

Corresponding Author(s): Eric Martens, U Michigan

Review Timeline:

Submission Date:	July 19, 2021
Editorial Decision:	August 25, 2021
Revision Received:	November 29, 2021
Accepted:	January 4, 2022

Editor: Mariana Byndloss

Reviewer(s): The reviewers have opted to remain anonymous.

Transaction Report:

DOI: <https://doi.org/10.1128/mSystems.00947-21>

August 25, 2021

Dr. Eric Martens
U Michigan
1150 West Medical Center Drive
Ann Arbor, MI 48109

Re: mSystems00947-21 (Phenotypic and genomic diversification in complex carbohydrate degrading human gut bacteria)

Dear Dr. Eric Martens:

Thank you for submitting your manuscript to mSystems. We have completed our review and I am pleased to inform you that, in principle, we expect to accept it for publication in mSystems. However, acceptance will not be final until you have adequately addressed the reviewer comments. Please make sure to include an "Importance" section in your revised manuscript. Also, we ask that you limit the number of supplementary materials to no more than 10 individual supplemental items, per mSystems guidelines.

Preparing Revision Guidelines

Sincerely,

Mariana Byndloss

Editor, mSystems

Journals Department
Reviewer comments:

Reviewer #1 (Comments for the Author):

This study is a tour-de-force where 354 gut Bacteroidales strains were each tested for growth in 45 different carbohydrate sources representative of those found in plant and animal dietary sources as well as host O-glycans. Nowhere has such a comprehensive analysis been published and it provides a unique overview of the nutritional sources and utilization diversity of this order of gut bacteria. There is a tremendous amount of data and numerous analyses of these data that will be extremely valuable to those who delve into the tables, figures and supplemental information, providing a tremendous asset to the field. The first sentence of the results section is a fantastic statement as to why careful phenotyping is essential. Indeed, their analyses prove that genomic analyses are not often reliable in predicting the polysaccharide sources that support the growth of various strains. The data show that although there is diversity within a species, strains of a given species cluster, suggesting that each species have distinct nutritional niches. The authors identified a negative correlation between the ability to utilize polysaccharides and endogenous mucin glycans and provide evidence that the ability to utilize mucin glycans is being lost in some species. I offer only a few suggestions for minor alterations to the text.

Minor suggestions

It is assumed that although likely originating from the gut, some of the VPI isolates may be clinical isolates from non-fecal samples. If the origin of the VPI isolates from the 60s and 70s is known or easily obtained, it would be helpful to include a column in Table S1.

Lines 93-94 - This statement should not be limited to industrialized countries at the phylum level. Humans from non-industrialized countries also have predominant Bacteroidetes species, but just different (*Prevotella* and *Alistipes*) than those in industrialized countries, however, all are Bacteroidetes. This statement would be true at the genus level of Bacteroides.

Line 97 - This statement is not specific enough. To state that you analyzed members of 29 different Bacteroidetes species is not incorrect, however, all species tested are those that have the ability to colonize the human colon, even if that was not the isolation source, and all are members of the order Bacteroidales. This study did not include environmental Flavobacteriales or Sphingobacteriales, or oral or vaginal Bacteroidales species, nor should it have based on the goals of the study. Please rewrite this statement to state that you tested 29 Bacteroidales strains that are known to colonize the human intestine, even if the source was animal or a human clinical isolate.

The last portion of the results section becomes somewhat difficult where the authors analyze the genome events leading to the pangenome evolution of *B. ovatus* and *B. xylanisolvens*. However, the methods and procedures for this analysis seem sound and the conclusions that are drawn match genetic and phenotypic data and likely serves as a more general example of LGT in the Bacteroides genus.

Discussion - line 470 - 471 - This sentence is confusing as written as this study did not offer new "experimental" support for genome exchange mechanisms. Maybe "bioinformatics" or genomic analysis.

Reviewer #2 (Comments to the Author attached to this email):

Reviewer #3 (Comments for the Author):

Pudlo et al. performed a high throughput growth analysis of Bacteroides species, represented by 352 different human and animal strains, against a wide range of complex carbohydrates coming from dietary, microbial, and host origins. The goal of the study is to link genomic data from gut microbiota to phenotypic functions of these commensal bacteria in their local intestinal niche. The results of this analysis showed that starch and fructans are ubiquitously utilised, whereas other polysaccharides are consumed in a species-specific manner. It also demonstrated that phylogenetically distant species can display similar utilisation profiles, indicating that carbohydrate metabolism is not strictly dictated by heritable traits. Interestingly, *B. fragilis* exhibited exclusive specificity for mucin O-glycans, utilisation of which negatively correlated with metabolism of dietary fibre. These observations support the notion that gut species have evolved to recognise and respond to carbohydrates that reflect their physiological niche.

Furthermore, the authors investigated whether genetic clusters mediating the breakdown of complex carbohydrates are conserved in closely related species. The authors focused on O-glycan utilisation in *B. ovatus* and *B. xylanisolvens*, splitting them into O-glycan degraders and non-degraders. It was found that more than half of the total genetic repertoire was not shared between the two species. Moreover, O-glycan degraders possessed a unique set of genes which were not shared between the members of this group. These observations suggests that *B. ovatus* and *B. xylanisolvens* strains are in the process of losing their ability to degrade mucin O-glycans. Lastly, the authors investigated whether some of the PULs or individual genes were exchanged between *B. xylanisolvens* and *B. ovatus* via homologous recombination. The analysis identified that a PUL previously characterised to implement β -mannan degradation in *B. ovatus* was transferred from a *B. xylanisolvens* ancestor. However, the high throughput growth analysis revealed that some *B. ovatus* strains lacking this PUL were still able to grow on β -mannan. This led to the identification of an additional PUL which potentially confers ability to utilise β -mannan in other *B. ovatus*.

Overall, the study demonstrates that *Bacteroides* have undergone an extensive genomic diversification driven by multiple genetic mechanisms. This study provides a rich resource to assess phenotypic abilities of gut *Bacteroides* to utilise complex and simple carbohydrates. Moreover, transcriptional data generated here could serve as a basis to elucidate mechanistic insights into mucus breakdown by gut *Bacteroides*, helping further identify their microbiological role in this process. While I remain enthusiastic with this study, several key concerns arose and need to be addressed:

Lines 170 - 176: state that *B. fragilis* and *B. vulgatus/B. dorei* display similar utilisation phenotypes, Fig 2 does not support this statement. Figure 2 shows that *B. vulgatus/B. dorei* preferentially degrade pectins (arabinan and RG1) and some strains can degrade mucin O-glycans, but the phenotype is not comparable to the one seen in *B. fragilis*.

Lines 191 - 195 This should reference figure S5 as well as figure S1. Figure S5 does not show a strong positive correlation between GAGs, pectins and all hemicelluloses, only monocot. Figure S5 also shows a positive correlation between GAGs, pectins, and microbial (fungal?) polysaccharides: dextran and α -mannan, not mentioned anywhere?

Lines 219 - 234: I am not sure how *Barnesiella intestinihominis* information is relevant here, also maybe could add *Barnesiella* and *B. masseliensis* phenotypes to Figure 2 or Figure S3 to demonstrate this difference in glycan preference. Seems like this section refers to the supplementary table 1 (which is a big dataset) and not backed up by any data in the main figures.

Figure 3 is misleading: *B. ovatus* circles do not correspond to the ones shown in the legend, the phenotype of *B. ovatus* H59 strains doesn't correspond to what is shown Figure 2 or Figure S3, especially if growths of *B. masseliensis* and *B. thetaiotaomicron* are used for comparison. Maybe there should be three groups: non-degraders, poor-degraders, and users.

Figure 6: Panel b is confusing, refers to lines 379 and 385. Looking at this figure I can't understand what the conclusion should be.

Lines 399 - 424 Summary: β -mannan PUL-A was transferred into *B.o* from a *B.x* ancestor. Despite lacking PUL-A, some *B.o* strains are still able to grow on β -mannan. This phenotype is mediated by β -mannan PUL-B which shares structural synteny with PUL-A. Figure 6c shows that *B.o* strains either retain PUL-A or PUL-B but not both. The origin of PUL-B is unclear and not discussed by the authors. It is also not discussed why some species possess PUL-A and not PUL-B and vice versa, this made this section seem unfinished.

The diagram of the PULs and qPCR of PUL-B should be presented in the main text.

As discussed and demonstrated in the O-glycan utilisation section, the presence of a PUL does not always translate into the phenotype. qPCR data shows that PUL-B is expressed but it doesn't demonstrate that it orchestrates β -mannan degradation. Figure 6c shows that some *B.x* and *B.o* strains do not have either of the PULs but are still able to grow on β -mannan, suggesting the presence of other putative PULs. So, what if the PUL-B is lost and the strains are still able to grow?

Figure S5 requires a legend explaining the colour coding and the cut-off for positive correlation.

Figure S8 shows a lot of data and doesn't have a figure legend.

Minor Corrections:

Figure S1:

Pectins: arabinose is a green star and rhamnose is a green triangle, makes it more visual. Also don't know if the side chain arabinose is α -1,6-linked. I thought that laminarin was β -1,3 backbone with β -1,6 sidechains, not mixed linkage?

Figure S5 lacks a label

Line 457: typo - should be: could then occur

Line 525: typo - Ot - should be 'at'

In the manuscript titled “Phenotypic and genomic diversification in complex carbohydrate degrading human gut bacteria” by Pudlo et al., the authors performed a large-scale phenotyping array to determine carbohydrate utilization profiles for over 350 members of Bacteroides, revealing wide variation in abilities of these bacteria to degrade substrates. In addition, Pudlo et al. performed extensive bioinformatic analysis to understand the connections between the presence of PULs, substrate-degrading traits, and nutritional niche specification (specialization) among the Bacteroides isolates. The authors also provided evidence for the potential LGT events that contribute to the remarkable mosaic in genomic architectures and variabilities in using mucin. Further, the authors also cleverly leveraged the transcriptomic responses to mucin to shed light on the evolutionary processes that shape mucin-utilization in *B. ovatus* and *B. xylanisolvens*. Overall, this excellent manuscript is well-written, and the experiments were well thought-out and executed. This paper not only revealed the incredible complexity of carbohydrate metabolism in the microbiota and the genomic events that potentially shape these phenomena but also provide an invaluable resource for the Bacteroides research community to understand the complex nutritional interactions between the microbes and the host in the gut.

Minor comments:

1. Was the large phenotypic assay done within a day? If not, can you authors provide information regarding how to normalize batch-to-batch difference (such as the incorporation of positive and negative controls in each batch)? If batch-to-batch normalization was not performed, please state that in the manuscript.
2. In the experiment described lines 177-190, the authors sought to test whether gut Bacteroides could simultaneously metabolize co-occurring polysaccharides. It is not clear whether the correlations between polysaccharide similarity and the abilities of Bacteroides to use these polysaccharides were performed in one single strain (*B. theta*) or all the Bacteroides strains included in this study. Logic dictates that the latter is the case, but this section could benefit from further clarification.
3. The experiment described between Line 372-391 is quite complex. Will it be simpler to compare the inter- vs. intraspecies median distance of the accessory gene sequences flanking all the genomic nodes?

Reviewer #1 (Comments for the Author):

This study is a tour-de-force where 354 gut Bacteroidales strains were each tested for growth in 45 different carbohydrate sources representative of those found in plant and animal dietary sources as well as host O-glycans. Nowhere has such a comprehensive analysis been published and it provides a unique overview of the nutritional sources and utilization diversity of this order of gut bacteria. There is a tremendous amount of data and numerous analyses of these data that will be extremely valuable to those who delve into the tables, figures and supplemental information, providing a tremendous asset to the field. The first sentence of the results section is a fantastic statement as to why careful phenotyping is essential. Indeed, their analyses prove that genomic analyses are not often reliable in predicting the polysaccharide sources that support the growth of various strains. The data show that although there is diversity within a species, strains of a given species cluster, suggesting that each species have distinct nutritional niches. The authors identified a negative correlation between the ability to utilize polysaccharides and endogenous mucin glycans and provide evidence that the ability to utilize mucin glycans is being lost in some species. I offer only a few suggestions for minor alterations to the text.

Minor suggestions

It is assumed that although likely originating from the gut, some of the VPI isolates may be clinical isolates from non-fecal samples. If the origin of the VPI isolates from the 60s and 70s is known or easily obtained, it would be helpful to include a column in Table S1.

Author response: We agree this would be useful metadata to report if we had it. Unfortunately, this was not obtained from the Salyers lab when we copied the strain collection.

Lines 93-94 - This statement should not be limited to industrialized countries at the phylum level. Humans from non-industrialized countries also have predominant Bacteroidetes species, but just different (*Prevotella* and *Alistipes*) than those in industrialized countries, however, all are Bacteroidetes. This statement would be true at the genus level of Bacteroides.

Author response: Thank you for catching this mistake. We indeed intended for this to reference members of the Bacteroides genus being highly abundant in industrialized population. We have re-worded to read:

“Members of the Bacteroidetes phylum are often among the most numerous bacteria in the human colonic microbiota, with members of the genus *Bacteroides* often prominent in individuals from industrialized countries (19-21).”

Line 97 - This statement is not specific enough. To state that you analyzed members of 29 different Bacteroidetes species is not incorrect, however, all species tested are those that have the ability to colonize the human colon, even if that was not the isolation source, and all are members of the order Bacteroidales. This study did not include environmental Flavobacteriales or Sphingobacteriales, or oral or vaginal Bacteroidales species, nor should it have based on the goals of the study. Please rewrite this statement to state that you tested 29 Bacteroidales strains that are known to colonize the human intestine, even if the source was animal or a human clinical isolate.

Author response: We have changed this to “29 Bacteroidales” as suggested.

The last portion of the results section becomes somewhat difficult where the authors analyze the genome events leading to the pangenome evolution of *B. ovatus* and *B. xylanisolvens*. However, the methods and procedures for this analysis seem sound and the conclusions that are drawn match genetic and phenotypic data and likely serves as a more general example of LGT in the *Bacteroides* genus.

Author response: We agree that this section gets complex, both in terms of the concept and visualization. In our revision, we have re-edited this section to attempt to make it clearer and also added some call outs to the final figure to make it more obvious which genes in the genomic analysis are candidates for LGT (e.g., Fig. 6B).

Discussion - line 470 - 471 - This sentence is confusing as written as this study did not offer new "experimental" support for genome exchange mechanisms. Maybe "bioinformatics" or genomic analysis.

Author response: Agreed. We have changed to “bioinformatics” support.

Reviewer #2 (Comments to the Author attached to this email):

In the manuscript titled “Phenotypic and genomic diversification in complex carbohydrate degrading human gut bacteria” by Pudlo et al., the authors performed a large-scale phenotyping array to determine carbohydrate utilization profiles for over 350 members of *Bacteroides*, revealing wide variation in abilities of these bacteria to degrade substrates. In addition, Pudlo et al. performed extensive bioinformatic analysis to understand the connections between the presence of PULs, substrate-degrading traits, and nutritional niche specification (specialization) among the *Bacteroides* isolates. The authors also provided evidence for the potential LGT events that contribute to the remarkable mosaic in genomic architectures and variabilities in using mucin. Further, the authors also cleverly leveraged the transcriptomic responses to mucin to shed light on the evolutionary processes that shape mucin-utilization in *B. ovatus* and *B. xylanisolvens*. Overall, this excellent manuscript is well-written, and the experiments were well thought-out and executed. This paper not only revealed the incredible complexity of carbohydrate metabolism in the microbiota and the genomic events that potentially shape these phenomena but also provide an invaluable resource for the *Bacteroides* research community to understand the complex nutritional interactions between the microbes and the host in the gut.

Minor comments:

1. Was the large phenotypic assay done within a day? If not, can you authors provide information regarding how to normalize batch-to-batch difference (such as the incorporation of positive and negative controls in each batch)? If batch-to-batch normalization was not performed, please state that in the manuscript.

Author response: Unfortunately, the experiment was not done in a single day (or even a single year for that matter! Table S1 lists the years each assay was run as part of the file names and all growth curve data is now provided as Source Data). Every plate contained two negative controls (water only, i.e. no carbohydrate added). These were checked manually and were nearly always zero growth in the medium formulations used. The only exceptions were some *Parabacteroides*, which showed slight background growth in the medium formulation (medium 2) that was used for them and contained 0.5 mg/ml beef extract. In these cases, the water blank growth was subtracted from total growth, although it's very difficult to subtract out a potential contribution to the "rate" component, if there is any, that might be driven by growth on background medium components. However, since growth was also generally poor on the medium with water only, the growth rate was nearly always slower than growth in carbohydrates. We've noted both of these details in the Methods, but otherwise do not provide additional description beyond the normalization scheme already used. The normalization should even out batch-to-batch differences along with the often larger strain-to-strain and species-to-species differences, which was our motivation for processing the data this way. However, we were not comfortable with normalizing any of the data without also providing the raw values in Table S1. In the revised manuscript, we have also provide the corresponding growth files (named according to the "file names" in column H in Table S1, which also have the year the analysis was run) as Source Data with the manuscript and will also provide a link to this on our website.

Added to Methods for phenotype array (Lines 548-553):

"Growth in the absence of an added carbohydrate source was generally not observed or very low, except with *Parabacteroides* that were often able to grow to a low level on the added 0.5% beef extract. The corresponding negative control wells for each strain assayed were averaged and this value subtracted from the total growth calculation of the corresponding to strain on other carbohydrates tested (All growth files are available at <https://www.ericmartenslab.org/>)."

2. In the experiment described lines 177-190, the authors sought to test whether gut *Bacteroides* could simultaneously metabolize co-occurring polysaccharides. It is not clear whether the correlations between polysaccharide similarity and the abilities of *Bacteroides* to use these polysaccharides were performed in one single strain (*B. theta*) or all the *Bacteroides* strains included in this study. Logic dictates that the latter is the case, but this section could benefit from further clarification.

Author response: The latter interpretation is correct. We have clarified this section to read as follows:

"To test for co-occurrence of different polysaccharide utilization abilities within the 354 individual strains, we calculated the pairwise correlations between utilization of any two polysaccharides by the same strain (Fig. S5). This might reveal tendencies to co-utilize different polysaccharides that are chemically different (positive correlation) or avoid using substrates from incompatible niches (negative correlation), if they exist."

3. The experiment described between Line 372-391 is quite complex. Will it be simpler to compare the inter- vs. intraspecies median distance of the accessory gene sequences flanking all the genomic nodes?

Author response: We agree this is complex, both to conduct and explain. As noted above, we have attempted to make this section clearer with regards to both how the analysis was conducted and how we explain the results. The effect of comparing inter- vs. intraspecies distances is not clear, but we don't think it would likely be simpler, at least from the perspective that 3 trees would need to be made for each gene instead of 1 (one each for the intra-species comparison, and another for the inter-species comparison). Looking for genes that are very distant from just the individual species median (*i.e.*, performing two intraspecies analyses) might provide extra sensitivity since the median for each species might in some cases (perhaps most?) be more characteristic of that species than a combination of both. It might, however, also identify genes that were transferred from another species if larger "distance from self" is the only determinant. Identifying new and more precise ways to determine these LGT events between multiple species will definitely be a goal for future work.

Reviewer #3 (Comments for the Author):

Pudlo et al. performed a high throughput growth analysis of *Bacteroides* species, represented by 352 different human and animal strains, against a wide range of complex carbohydrates coming from dietary, microbial, and host origins. The goal of the study is to link genomic data from gut microbiota to phenotypic functions of these commensal bacteria in their local intestinal niche. The results of this analysis showed that starch and fructans are ubiquitously utilised, whereas other polysaccharides are consumed in a species-specific manner. It also demonstrated that phylogenetically distant species can display similar utilisation profiles, indicating that carbohydrate metabolism is not strictly dictated by heritable traits. Interestingly, *B. fragilis* exhibited exclusive specificity for mucin O-glycans, utilisation of which negatively correlated with metabolism of dietary fibre. These observations support the notion that gut species have evolved to recognise and respond to carbohydrates that reflect their physiological niche.

Furthermore, the authors investigated whether genetic clusters mediating the breakdown of complex carbohydrates are conserved in closely related species. The authors focused on O-glycan utilisation in *B. ovatus* and *B. xylanisolvens*, splitting them into O-glycan degraders and non-degraders. It was found that more than half of the total genetic repertoire was not shared between the two species. Moreover, O-glycan degraders possessed a unique set of genes which were not shared between the members of this group. These observations suggests that *B. ovatus* and *B. xylanisolvens* strains are in the process of losing their ability to degrade mucin O-glycans. Lastly, the authors investigated whether some of the PULs or individual genes were exchanged between *B. xylanisolvens* and *B. ovatus* via homologous recombination. The analysis identified that a PUL previously characterised to implement β -mannan degradation in *B. ovatus* was transferred from a *B. xylanisolvens* ancestor. However, the high throughput growth analysis revealed that some *B. ovatus* strains lacking this PUL were still able to grow on β -mannan. This led to the identification of an additional PUL which potentially confers ability to utilise β -

mannan in other *B. ovatus*.

Overall, the study demonstrates that *Bacteroides* have undergone an extensive genomic diversification driven by multiple genetic mechanisms. This study provides a rich resource to assess phenotypic abilities of gut *Bacteroides* to utilise complex and simple carbohydrates. Moreover, transcriptional data generated here could serve as a basis to elucidate mechanistic insights into mucus breakdown by gut *Bacteroides*, helping further identify their microbiological role in this process. While I remain enthusiastic with this study, several key concerns arose and need to be addressed:

Lines 170 - 176: state that *B. fragilis* and *B. vulgatus/B. dorei* display similar utilisation phenotypes, Fig 2 does not support this statement. Figure 2 shows that *B. vulgatus/B. dorei* preferentially degrade pectins (arabinan and RG1) and some strains can degrade mucin O-glycans, but the phenotype is not comparable to the one seen in *B. fragilis*.

Author response: We can see where the “absolute” nature of this statement, especially with regard to use of the term “very similar”, was confusing as originally written. Indeed, in the original description, the sentence that followed the one at Line 170 noted the presence of pectin utilization (often quite weak, except for RG1 and arabinan) in this lineage in an attempt highlight this difference among otherwise more specialized organisms. We have expanded the description and now call out the additional abilities in *Bv/Bd*, which might allow them to diverge from competitors with otherwise similar phenotypes. This section now reads:

“Despite being phylogenetically more distant, members of these two species possess similar abilities to degrade starch and related molecules (glycogen, pullulan), inulin and mucin O-glycans. The major distinguishing feature between these groups is the presence of some pectin utilization, which is often weak, among strains of *B. vulgatus/dorei*. Indeed, acquisition of growth abilities that are unique with respect to species with otherwise similar potential may be one way that species avoid direct competition for the same niches.”

Lines 191 - 195 This should reference figure S5 as well as figure S1. Figure S5 does not show a strong positive correlation between GAGs, pectins and all hemicelluloses, only monocot. Figure S5 also shows a positive correlation between GAGs, pectins, and microbial (fungal?) polysaccharides: dextran and α -mannan, not mentioned anywhere?

Author response: We apologize for the oversight with respect to calling out figures. This section should really only reference Fig. S5 as noted. There may be some confusion about the correlation between utilization of individual polysaccharides in the categories GAGs and hemicelluloses. It was not meant to imply that these groups were correlated but individual polysaccharides *within* the GAG group are correlated (this makes sense since *B. theta* utilizes 3 of the 4 GAGs tested through a single PUL) and the same for the pectins (although as correctly pointed out there is a correlation between the monocot hemis and some GAGs/pectins. We have chosen to just highlight the within group correlations in this section and have re-worded this section to read:

“We also observed positive correlations in the ability of bacteria to simultaneously utilize polysaccharides within two different groups of plant cell wall polysaccharides (pectins and hemicelluloses), as well as

animal tissue glycosaminoglycans (Fig. S5, green boxes highlight the 3 separate groups containing substrates with positive correlations within that group, although weaker correlation can be observed across groups). These correlations occurred despite the fact that the polysaccharides within each of these groups often possess different structures but might co-occur in plant material or digested animal tissue.”

We didn’t originally call out dextran and α -mannan since we don’t know of any intrinsic co-occurrence of those polysaccharides in the same food, organisms, etc. But, we have added a statement noting the correlation despite a lack co-occurrence in foods to other sources.

“Finally, there was a positive correlation between utilization of α -mannan and dextran, two microbial polysaccharides that are not known to occur together in foods or other sources of these polysaccharides (Fig. S5).”

Lines 219 - 234: I am not sure how *Barnesiella intestinihominis* information is relevant here, also maybe could add *Barnesiella* and *B. massiliensis* phenotypes to Figure 2 or Figure S3 to demonstrate this difference in glycan preference. Seems like this section refers to the supplementary table 1 (which is a big dataset) and not backed up by any data in the main figures.

Author response: The data for these species is actually shown in Fig. 2. However, since there are only 1 or 3 isolates of species, respectively, the group was too small to label or stand out clearly in the original figure. This was part of the motivation for describing these specialists in the text, so the reader doesn’t need to scrutinize Table S1. To better call these out and since the 4 isolates all cluster in the same region as would be expected based on their similar specialization, we have annotated this region in Fig. 2. The following was added to the Figure 2 legend:

“The region containing mucin specialists *B. massiliensis* and *Ba. intestinihominis* is indicated but marked with an asterisk because the 4 strains in these two species are not perfectly clustered in this region.”

Figure 3 is misleading: *B. ovatus* circles do to not correspond to the ones shown in the legend, the phenotype of *B. ovatus* H59 strains doesn't correspond to what is shown Figure 2 or Figure S3, especially if growths of *B. massiliensis* and *B. thetaiotaomicron* are used for comparison. Maybe there should be three groups: non-degraders, poor-degraders, and users.

Author response: We may be missing an intended point about the legend to figure 3A, but the numbers shown in the legend are just some reference circle sizes (58 was the maximum since that was the deepest species sample and the circles are basically half sizes from there on down). We’ve checked the size of the *B. ovatus* sample (black circle) and it corresponds to the actual sample size (33 strains). The interior circle denoting strains that surpass our threshold for mucin degradation (red) is also accurate (26 strains). We feel that this figure is useful in first introducing the prevalence of mucin degradation across the phylogeny of species tested (this information is lacking in Fig. 2). Then, Figure 3B shows a more granular view of just *B. ovatus* and *B. xylanisolvens*. The point about O-glycan utilization being a more continuous phenotype rather than present/absent is a very good one and we struggled with how to best describe this as well. In the end, we decided to use a cohesive (albeit binary) scheme for calling growth

on any substrate based on a uniform (and rather low) threshold (*i.e.*, growth > 0.1 Abs₆₀₀ units). Then elaborating on the case of mucin glycans in more detail. We've added a label to Fig. 3A to note the Bo/Bx lineage in which mucin utilization is variable in presence and strength. We've also added some additional description in this section to note that continuous nature of this phenotype, which is likely due in part to the genetic complexity that underlies it (e.g., strains with partial PUL repertoires):

[Lines 285-289]. "Among the Bo and Bx strains that surpassed the threshold for growth on O-glycans there was a continuous gradient of growth abilities, which could be attributed to variations in PUL content and therefore gradations in the strains' abilities to access the many different structures in the complex O-glycan mixture (Fig. 3B)."

After becoming confused ourselves upon checking the strain names in Fig. S3, we believe we realized the point of confusion. The labels in Fig. S3 are very small and need to be zoomed in to see. However, the rightmost column in the Fig. S3 heatmap is the last monosaccharide tested (xylose) and not mucin O-glycans. To clarify this, we have copied the list of names and placed them at the right of the polysaccharide heatmap block too, which places them right next to the O-glycan column. Using this, we have carefully checked to make sure that the strong mucin degraders show up as positive signals in the heatmap.

Figure 6: Panel b is confusing, refers to lines 379 and 385. Looking at this figure I can't understand what the conclusion should be.

Author response: This is an area where we attempted to improve clarity. Basically, the blue dots that are high on the y axis and the red dots that are far right on the x axis indicate the alleles that are far away from those in the "self" species. We've added extra labels on the figure to point out the take away, which we think draws the reader to the main conclusion.

Lines 399 - 424 Summary: β -mannan PUL-A was transferred into B.o from a B.x ancestor. Despite lacking PUL-A, some B.o strains are still able to grow on β -mannan. This phenotype is mediated by β -mannan PUL-B which shares structural synteny with PUL-A. Figure 6c shows that B.o strains either retain PUL-A or PUL-B but not both. The origin of PUL-B is unclear and not discussed by the authors. It is also not discussed why some species possess PUL-A and not PUL-B and vice versa, this made this section seem unfinished.

Author response: Essentially, we think that PUL-A and PUL-B are non-orthologous PULs that confer the same ability to grow on galactomannan. If a strain has either system or both it is capable of growth on this branched mannan. Indeed, we don't really know the true "origin" of any PUL that is observed to be present in an extant Bacteroidetes strain. But, based on the prevalence data, it would seem that PUL-A is a GalMan utilization system that is more prevalent and perhaps originated in *B. xylanisolvens*, it is also capable of transfer to *B. ovatus*. PUL-B is more prevalent in *B. ovatus* and may have origins in that species, at least with respect to *B. xylanisolvens* where it has so far not been observed. While we've shown that PUL-B is highly expressed during growth on GalMan (this has always been a strong indicator of

involvement in that phenotype, but incomplete proof of function) we would either need to identify a genetically tractable strain with only PUL-B and delete it or perform thorough enzymatic analysis of the GHs it encodes. The other open question in this section is the source of the residual GluMan growth in strains lacking both PUL-A and PUL-B. We attempted to resolve this by searching for GH26 containing PULs in those strains and, while multiple candidates can be found, the connection is unclear. One of the common GH26 containing candidate PULs is actually present in *B. ovatus* 8483, for which we have previously shown that only PUL-A is required for growth on both GalMan and GluMan, so it is unclear if it plays any role in GluMan utilization. If it makes the take home message of Fig. 6C more clear (*i.e.*, that non-orthologous PULs exist for the same function and at least one is part of the LGT pool), we could remove the GluMan data from this figure and focus just on GalMan for which the presence of either PUL-A and/or PUL-B is a perfect correlation.

To flesh the idea above out a bit more in the paper, we have added the following text to the Discussion.

[lines 497-501] “Based on the prevalence data, it seems that PUL-A is a GalMan utilization system that is more prevalent in, and perhaps also originated in, *Bx* and it is also capable of transfer to *Bo*. PUL-B is more prevalent in *Bo* and may have origins in that species, at least with respect to *Bx* where it has so far not been observed.”

The diagram of the PULs and qPCR of PUL-B should be presented in the main text.

As discussed and demonstrated in the O-glycan utilisation section, the presence of a PUL does not always translate into the phenotype. qPCR data shows that PUL-B is expressed but it doesn't demonstrate that it orchestrates β -mannan degradation. Figure 6c shows that some *B.x* and *B.o* strains do not have either of the PULs but are still able to grow on β -mannan, suggesting the presence of other putative PULs. So, what if the PUL-B is lost and the strains are still able to grow?

Author response: As discussed above, the presence of PUL-A and/or PUL-B is perfectly correlated with the ability to grow on GalMan, although we agree that we fall short of confirming the role of PUL-B with genetics or enzyme studies. We have not found any strains that do not have PUL-A/B and can grow on GalMan (as noted above the low residual GluMan growth is an open question).

Figures S5 requires a legend explaining the colour coding and the cut-off for positive correlation.

Author response: This has been added to the figure legend (anything >0.4 in either positive or negative direction was highlighted with the gradation of color intensity shown in the key).

New text added to the Figure S5 legend:

“For each substrate pair, the values shown indicate the positive or negative correlation value that both substrates will be used by any of the strains among the 354 surveyed. Positive or negative correlations that are >0.40 are shown in the colors indicated.”

Figures S8 shows a lot of data and doesn't have a figure legend.

Author response: We provided a legend for this in the original version but agree that it was far too brief and was essentially just a title. The data shown here parallel the example discussed more deeply in Fig. 6C, so we made the unfair assumption that the reader would take it as additional examples of PUL and non-PUL LGT events. We have expanded this legend to introduce what is shown and discuss the examples.

Minor Corrections:

Figure S1:

Pectins: arabinose is a green star and rhamnose is a green triangle, makes it more visual. Also don't know if the side chain arabinose is α -1,6-linked. I thought that laminarin was β -1,3 backbone with β -1,6 sidechains, not mixed linkage?

Author response: Thank you for catching these older symbols. These symbol and linkage changes have been made. Also, thank you for catching the mistake in the laminarin linkages, which have also been corrected. We noted Ara- α 1,6 linkages in AG sidechains based on DOI10.1016/j.pbi.2008.03.006. Please advise if there is a more current model, as it quite possible that we missed a revision to the structural understanding of these pectic sidechains. The Megazyme website still shows branching α 1,4 Ara in larch arabinogalactan, albeit directly attached to the arabinogalactan chain.

Figure S5 lacks a label

Author response: We've added a label to this figure.

Line 457: typo - should be: could then occur

Author response: Corrected.

Line 525: typo - Ot - should be 'at'

Author response: Corrected.

January 4, 2022

Dr. Eric Martens
U Michigan
1150 West Medical Center Drive
Ann Arbor, MI 48109

Re: mSystems00947-21R1 (Phenotypic and genomic diversification in complex carbohydrate degrading human gut bacteria)

Dear Dr. Eric Martens:

I am happy to report that your manuscript has been accepted, and I am forwarding it to the ASM Journals Department for publication. For your reference, ASM Journals' address is given below. Before it can be scheduled for publication, your manuscript will be checked by the mSystems senior production editor, Ellie Ghatineh, to make sure that all elements meet the technical requirements for publication. She will contact you if anything needs to be revised before copyediting and production can begin. Otherwise, you will be notified when your proofs are ready to be viewed.

Publication Fees:

We recognize that the video files can become quite large, and so to avoid quality loss ASM suggests sending the video file via <https://www.wetransfer.com/>. When you have a final version of the video and the still ready to share, please send it to mssystemsjournal@msubmit.net.

Sincerely,

Mariana Byndloss
Editor, mSystems

Journals Department
Phone: 1-202-942-9338